psychology/human-computer interaction

dysregulated gaming, gaming disorder, specification curve analysis, well-being

**Author for correspondence:**
Nick Ballou
e-mail: nick.ballou@gmail.com

# The relationship between mental well-being and dysregulated gaming: a specification curve analysis of core and peripheral criteria in five gaming disorder scales

## Nick Ballou[1] and Antonius J. Van Rooij[2]

[1]Queen Mary University of London, London, UK
[2]Trimbos Institute, Utrecht, The Netherlands

(ID) NB, 0000-0003-4126-0696

Gaming disorder (also known as dysregulated gaming) has received significant research and policy attention based on concerns that certain patterns of play are associated with decreased mental well-being and/or functional impairment. In this study, we use specification curve analysis to examine analytical flexibility and the strength of the relationship between dysregulated gaming and well-being in the form of general mental health, depressive mood and life satisfaction. Dutch and Flemish gamers ($n = 424$) completed an online survey containing five unique dysregulated gaming measures (covering nine scale variants) and three well-being measures. We find a consistent negative relationship; across 972 justifiable regression models, the median standardized regression coefficient was −0.39 (min: −0.54, max: −0.19). Data show that the majority of dysregulated gaming operationalizations converge upon highly similar estimates of well-being. However, variance is introduced by the choice of well-being measure; results indicate that dysregulated gaming is more strongly associated with depressive mood than with life satisfaction. Weekly game time accounted for little to no unique variance in well-being in the sample. We argue that research on this topic should compare a broad range of psychosocial well-being outcomes and explore possible simplifications of the DSM-5 gaming disorder criteria. Given somewhat minute differences between dysregulated gaming scales when used in survey-based studies and largely equivalent relationships with mental health indicators, harmonization of measurement should be a priority.

# 1. Introduction

Some players of video games develop a pattern of behaviour in which video game use becomes dysregulated, accompanied by significant psychological distress and/or functional impairment. This phenomenon is colloquially referred to as 'gaming addiction,' but is also known as (Internet) gaming disorder (IGD), dysregulated gaming, or problematic video game use. In the current work, we adopt the term dysregulated gaming; indicators of dysregulation are self-reported by respondents, but we are hesitant to assess a clinical disorder on the basis of survey data alone at this stage [1].

Over the past decades, worldwide recognition that patients sometimes present with dysregulated gaming issues has stimulated efforts to formalize the issue as a psychopathological condition in manuals. This has prominently led to the inclusion of IGD as a condition warranting further study in the fifth edition of the Diagnostic and Statistical Manual of Mental Disorders (DSM-5) [2], and the addition of gaming disorder to the ICD-11 framework [3]. However, the status of dysregulated gaming as a psychopathological condition remains heavily debated [4–6]. This includes areas of uncertainty regarding measurement, comorbidity, symptomology and more.

Concern about dysregulated gaming largely stems from the negative outcomes with which it is associated, rather than about engagement with gaming *per se*. As opposed to games being inherently harmful, as evidenced by the extensive literature on positive effects that can result from video game play [7], it is generally thought that certain patterns of game use may be either associated with or result in detrimental effects to other important aspects of the players' lives. Gentile *et al.* [8] write 'playing is not pathological initially but becomes pathological for some individuals when the activity becomes dysfunctional, harming the individual's social, occupational, family, school and psychological functioning' (p. e320). Negative outcomes that have been linked with dysregulated gaming include psychopathology (e.g. depression or anxiety) (e.g. [9]), decreased work/academic performance [10], poorer physical health (e.g. [11]), lower life satisfaction (e.g. [12]) and social problems (e.g. [13]).

Other work, however, has not always supported these links. Some researchers have failed to find a negative relationship between dysregulated gaming and well-being (e.g. [14]), whereas others have interpreted effects that they have found to be too small to be meaningful [15]. Even when negative effects are found, the causal nature of the relationship has been heavily debated, with some evidence suggesting that dysregulated gaming may be better understood as a symptom of an underlying disorder or disorders [6]. In sum, more evidence is needed both with regard to the strength of the relationship between dysregulated gaming and well-being, and whether this varies between different aspects of well-being.

In this paper, we use specification curve analysis in an attempt to shed light on this relationship. We compare five distinct dysregulated gaming questionnaires, encompassing nine possible measurement approaches, alongside three different well-being measures, allowing us to investigate the influence of analytical flexibility on results. We further compare core and peripheral dysregulated gaming criteria [16] and the influence of covariates or 'risk factors' identified in previous literature, like amount of time spent gaming and gaming motivations.

Due to the wide scope of the analyses presented here, the key findings are summarized in Box 1 as a resource for the reader. Each of these findings will be examined in more detail throughout the Method, Results and Discussion sections.

## 1.1. Core versus peripheral criteria

One particularly contentious aspect of dysregulated gaming research concerns the differentiation of high-engagement, non-pathological gaming from true dysregulation. Dysregulated gaming criteria have often been adapted from other behavioural addictions, particularly problem gambling. Measures developed to assess dysregulated gaming, whether based directly on the DSM-5 or otherwise, regularly include both criteria that directly assess negative consequences of gaming on a person's well-being or functioning as well as criteria that ask about a person's cognitive and behavioural engagement, but that are not inherently negative [17].

This distinction has sometimes referred to as 'core' versus 'peripheral' criteria [16,18]. A large body of work has shown that high engagement with games does not imply negative consequences, even for players who play dozens of hours per week. This distinction has been made on theoretical grounds, using evidence from case studies [19], latent class analysis [20], and survey work showing weaker associations between psychological risk factors (e.g. [21]) and functional outcomes (e.g, [22]) for highly engaged gaming than for dysregulated gaming.

> **Box 1.** Summary of key findings.
>
> — Dysregulated gaming scales were more similar than they were different in terms of their association with well-being, with eight of nine measures converging upon highly similar effect size estimates
>
> — Dysregulated gaming was more strongly linked with increased depressive mood than with lower life satisfaction
>
> — A peripheral DSM-5 criterion, escapism, was the strongest single-item predictor of decreased well-being; however, it may be confounded and assess the presence of problems in a player's daily life, rather than solely their use of games to escape problems
>
> > — The same may be true to a slightly lesser extent of escapism as a player motivation
>
> — Although escapism was strongly predictive of well-being on its own, the inclusion of peripheral DSM-5 criteria did not meaningfully strengthen or obscure the relationship between dysregulated gaming and well-being compared with using core criteria only
>
> — In line with other work, no meaningful relationship was found between self-reported weekly playtime and well-being
>
> — While they remained similarly predictive of well-being, two previously used scales (VAT and CVAT-2) were not found to be unidimensional in our sample
>
> > — Psychometric evidence supported the validity of five other scale variants

Early work on dysregulated computer use used factor analysis to show that the tolerance, euphoria and preoccupation (cognitive salience) criteria loaded onto one separate factor, referred to as 'engagement', while inter- and intra-personal conflict, relapse and reinstatement, withdrawal, and loss of opportunities loaded onto a second factor, referred to as 'addiction' [23], with this distinction later being replicated for games [16]. Much work has followed from this; [24], for example, use item response theory to show that two peripheral DSM-5 criteria, preoccupation and escapism, are poor at discriminating dysregulated gamers, and argue for their removal from dysregulated gaming measures.

Preoccupation has been identified as a particularly problematic peripheral item, with [25] finding that the IGD salience criterion reached an endorsement rate as high as 77.3% on a dichotomized scale in their sample. A recent review article [26] identified withdrawal, inter- and intra-personal conflict, unsuccessful attempts to stop (lapse/relapse), loss of control, and continuation despite problems as likely core criteria; preoccupation, tolerance, lying about use, and mood modification (escapism), on the other hand, were listed as likely peripheral criteria.

Despite general agreement that current scales conflate core and peripheral criteria, delineations of which criteria may be considered core or peripheral have not always been clear-cut. Deleuze *et al.* [25], for example, found that mood modification more closely resembled a core criterion. In this paper, we adopt a conservative position and only classify criteria directly assessing a negative consequence on the person's life (including internal psychological conflict) as a result of their gaming as core criteria (see appendix for item-wise classification).

## 1.2. Gaming motivations and basic psychological need satisfaction in relation to dysregulated gaming

### 1.2.1. Gaming motivations

Significant research attention has been paid to covariates of dysregulated gaming. In particular, the motivational styles of players have emerged as a popular topic of inquiry. Hilgard *et al.* [27], for example, found that players who scored highly on three types of motivation—escapism, social interaction and grinding—were more likely to exhibit symptoms of dysregulated gaming. Several studies have used versions of Yee's [28] Motives for Play in Online Games questionnaire, with one example finding that escapism ($\beta = 0.36$) and mechanics ($\beta = 0.21$) were linked with problematic video game use [29]. Billieux *et al.* [30] report related findings, showing that Yee's advancement, mechanics, competition, role-play, customization and escapism motives were linked with dysregulated gaming symptoms, and that those motives also predicted particular in-game behaviours and progression over time. Another study found that entertainment/achievement, social and escapism motives were

associated with problematic gaming ($\beta s = 0.19$, 0.23 and 0.30) [31]. Overall, researchers have identified a range of motives as possible risk factors for dysregulated gaming, with escapism noted as a particularly common one with conceptual implications given its overlap with a peripheral DSM-5 criterion. When studying the relationship between well-being and dysregulated gaming, it therefore seems wise to include these motivations in the analysis process to investigate their role as complementary or potentially overlapping constructs.

### 1.2.2. Self-determination theory and dysregulated gaming

Self-determination theory (SDT; [32,33]) has emerged as a popular framework for understanding motivation to play games. One of SDT's sub-theories, basic psychological needs theory, states that activities and environments are most likely to be motivating and support well-being when they provide for the satisfaction of three needs, namely autonomy (the need to feel that one's actions are volitional and self-endorsed), competence (the need to act effectively and experience growth) and relatedness (the need to have one's authentic self accepted by others and to feel both caring and cared for). A large body of work has found that satisfaction of basic needs in video games has been positively linked with enjoyment and motivation to play (e.g. [34,35]).

Further work has examined in-game need satisfaction in the context of dysregulation. Two studies have found that need satisfaction in games was associated with dysregulated gaming, but that both this relationship and the relationship between need satisfaction in games and well-being was moderated by need dissatisfaction/frustration in daily life [36,37]. Bender & Gentile [38] write that need satisfaction in games may operate as a risk factor, and need satisfaction in real life as a protective factor, with each operating somewhat independently. Given these findings, we include the experience of need satisfaction in video game contexts as a further covariate.

### 1.3. Present research

Research on games disorder currently makes use of a wide range of questionnaires, with a recent systematic review identifying 32 existing scales with varying degrees of empirical support [17]. However, scales are rarely directly compared against each other, leaving us with an incomplete understanding of the degree to which analytical flexibility might drive results when using dysregulated gaming as either an explanatory or outcome variable. When comparisons have occurred, they have typically been done in the context of convergent validity, or comparing a new dysregulated gaming measure with an existing one, and include only two scales (e.g. [39,40]).

In the present research, we extend this, focusing on the relationship between dysregulated gaming measures and mental well-being, namely general mental health, life satisfaction and depressive mood. We use specification curve analysis to integrate results from five dysregulated gaming measures (yielding nine possible specifications) and three well-being measures. We select and, where necessary, have developed measures to include (i) an adaptation of earlier compulsive Internet use scales, (ii) the DSM-5 Internet gaming disorder criteria, (iii) the ICD-11 gaming disorder criteria, (iv) criteria from a large-scale international project on children's digital well-being (Global Kids Online), and (v) a collaborative attempt to define behavioural addiction. Given that dysregulated gaming is expected to be linked with decreased well-being, this can also be framed as an investigation of the concurrent (criterion) validity of different dysregulated gaming scales. Our research questions are threefold:

1. What is the strength of the relationship between dysregulated gaming severity and psychological well-being?
2. How much variability in the size of that effect can be attributed to analytical choices, including the selection and operationalization of the criterion variable (dysregulated gaming severity), outcome variable (well-being), and the in/exclusion of various covariates (demographics, gaming motivations, and need satisfaction in games)?
3. Are dysregulated gaming core criteria more strongly associated with decreased mental well-being than engagement criteria?

# 2. Method

## 2.1. Procedure

Dutch and Flemish (Dutch-speaking Belgian) subjects were recruited between December 2018 to March 2019 to complete an online survey built in Limesurvey. Flemish participants were recruited from a gaming news website (4gamers.be), while Dutch participants were recruited through a Facebook ad campaign associated with a public health website (gameninfo.nl); a minority of Dutch participants (7%) joined the survey from the public health website directly. Respondents were required to be at least 16 years old and complete an informed consent form. In exchange for finishing the survey, participants were entered into a raffle for one of six €50 Gamegear gift certificates.

Participants completed Dutch language versions of the measures below, totalling 132 items. Measures were presented in a fixed order, with the order of items within each measure randomized. The survey sequencing was chosen to avoid the possibility of multiple consecutive negatively interpreted measures and/or monotony by alternating dysregulated gaming scales with non-dysregulated gaming scales (e.g. demographic questions, gaming motivations, or well-being measures). Longer measures were placed toward the beginning of the survey to mitigate survey fatigue. We included two attention checks and a number of reverse-coded items throughout the questionnaire.

## 2.2. Participants and cleaning

A total of 743 people began the survey, and after cleaning we retained 424. We removed 289 (38.9%) people who began the survey but did not complete it. Contrary to expectations, the presentation of multiple thematically similar measures does not appear to have been the driving force for this relatively high dropout rate; of the 491 participants who made it as far as the second dysregulated gaming measure, 454 completed the entire survey. Instead, 87% of the participants who abandoned the survey did so during the first three measures. While we cannot be sure of the cause of participant dropout, we suspect that the slow filling of the survey progression bar after each page signalled to participants that the questionnaire was lengthier than they had expected or were willing to complete.

We additionally removed 30 participants who entered an incorrect answer for one or both of the two careless response checks [41]. This decision was corroborated by manual inspection of the data, which revealed high rates of uniform or patterned responses among these. Overall, the quality of responses for participants who completed the full survey is high; only 6.6% of the participants who completed the survey were dropped based on the careless response check, and reverse-coded items performed as expected in all scales in which they appear. This resulted in our final sample size of 424.

Of the final participants, 220 were Dutch and the remainder were Flemish (204). The average age was 24.2 (s.d. = 7.7). This included 307 males, 113 females and four participants who preferred to specify another gender. The median completion time for the survey was 14 min. Each item required an answer, and therefore after removing incomplete survey responses according to the procedure above, there were no missing data.

A $t$-test revealed that Dutch participants reported higher dysregulated gaming on average ($t[502.77] = -2.50$, $p = 0.013$); this is probably a result of the fact that Dutch participants were significantly younger than Flemish ones ($t[378.60] = 3.41$, $p = 0.001$, $M = 20.8$ versus 27.9). Given overall high similarity between the two groups (same language, high degree of cultural overlap, participants all self-selected due to interest in gaming) and no theoretical reason to expect the relationship between dysregulation and well-being to vary between the two countries, we elected to analyse them together.

Further tests indicated two significant differences between participants dropped due to incomplete responses or careless response checks compared with the rest of the sample. Dropped participants reported on average lower educational levels ($\chi^2[1, n = 424] = 5.68$, $p = 0.017$) and higher scores on the first of the presented dysregulated gaming measure, the CVAT ($t[134.47] = 2.22$, $p = 0.028$). No statistically significant differences were detected for the other demographic variables, age and gender. As a result of these differences, we conducted a sensitivity analysis in which both incomplete and careless responses were included; this did not meaningfully change the pattern of results (see electronic supplementary material).

## 2.3. Measures

All measures were assessed on a five-point Likert scale where 1 is 'completely disagree,' 3 is 'neither disagree nor agree,' and 5 is 'completely agree.' Exact wordings for each measure can be found in the appendix. Scale scores were calculated by averaging the relevant items (after reverse coding, as necessary). In addition to the measures below, participants also reported their ambitions for digital

**Table 1.** Descriptive statistics for explanatory, criterion and control variables. VAT, Video Game Addiction Test; CVAT-2, Clinical Video Game Addiction Test 2.0; ODBA, open definition of behavioral addictions; GKO, Global Kids Online. $\omega_h$ = McDonald's hierarchical omega reliability coefficient, with point estimates and confidence intervals calculated using 1000 bias-corrected and accelerated bootstrap samples, as recommended by [42].

| | mean | s.d. | min | max | $\omega_h$ |
|---|---|---|---|---|---|
| **dysregulated gaming** | | | | | |
| VAT | 2.22 | 0.66 | 1.00 | 5.00 | (table 2) |
| CVAT-2 (9-item) | 2.09 | 0.68 | 1.00 | 4.11 | (table 2) |
| CVAT-2 (11-item) | 2.09 | 0.67 | 1.00 | 4.18 | (table 2) |
| ICD-11 | 1.82 | 0.79 | 1.00 | 5.00 | (table 2) |
| ODBA | 1.65 | 0.57 | 1.00 | 4.00 | (table 2) |
| GKO | 1.58 | 0.67 | 1.00 | 4.60 | (table 2) |
| CVAT-3 | 2.02 | 0.68 | 1.00 | 4.27 | (table 2) |
| **well-being** | | | | | |
| general mental health | 3.72 | 0.80 | 1.20 | 5.00 | 0.82 [0.78,0.85] |
| absence of depressive mood | 3.61 | 0.92 | 1.00 | 5.00 | 0.86 [0.84,0.88] |
| life satisfaction | 3.34 | 0.80 | 1.20 | 5.00 | 0.82 [0.79,0.85] |
| **demographics** | | | | | |
| age | 24.36 | 7.74 | 16.00 | 61.00 | n.a |
| game time per week (hours) | 23.42 | 14.80 | 2.00 | 90.00 | n.a |
| game session length (minutes) | 151 | 76 | 2 | 420 | n.a. |
| **covariates** | | | | | |
| autonomy satisfaction | 3.84 | 0.55 | 1.00 | 5.00 | 0.73 [0.66,0.79] |
| competence satisfaction | 3.94 | 0.58 | 1.00 | 5.00 | 0.81 [0.76,0.85] |
| relatedness satisfaction | 3.32 | 0.92 | 1.00 | 5.00 | 0.88 [0.85,0.90] |
| habit motivation | 3.73 | 0.87 | 1.00 | 5.00 | 0.83 [0.80,0.86] |
| narrative motivation | 4.02 | 0.64 | 1.00 | 5.00 | 0.71 [0.65,0.77] |
| pastime motivation | 3.06 | 0.93 | 1.00 | 5.00 | 0.82 [0.78,0.85] |
| performance motivation | 3.75 | 0.61 | 1.67 | 5.00 | 0.69 [0.63,0.74] |
| reactivity motivation | 4.28 | 0.56 | 1.67 | 5.00 | 0.66 [0.56,0.72] |
| social motivation | 3.35 | 0.94 | 1.00 | 5.00 | 0.80 [0.76,0.84] |
| escapism motivation | 3.92 | 0.66 | 1.33 | 5.00 | 0.61 [0.51,0.67] |
| agency motivation | 3.74 | 0.59 | 1.33 | 5.00 | 0.71 [0.64,0.76] |

careers (e.g. streamer or esports competitor), one or more favourite games and commonly used devices. This data are not analysed here.

Table 1 shows the descriptive statistics for each measure, and figure 1 displays zero-order correlations between all variables in the study. Descriptive statistics are reported based on the sample with outliers excluded (see analytical approach below). Cronbach's $\alpha$ is omitted in favour of McDonald's hierarchical omega ($\omega_h$) as a measure of reliability, as the latter does not assume essential tau-equivalence (i.e. equal factor loadings) and is known to be a more appropriate estimate [43,44]. Values of $\omega_h$ can be interpreted similarly to $\alpha$ values. Reliability for the dysregulated gaming measures is reported in the factor analytic results (table 2) below.

## 2.4. Well-being

### 2.4.1. General mental health

General mental health was measured with the five-item Mental Health Inventory (MHI), which constitutes the mental health sub-scale of the Medical Outcomes Study (MOS) Short-Form Health

Survey [45]. The MHI is well-validated, including in a Dutch sample [46], and has been used for the screening of major depression, dysthemia, as well as certain anxiety disorders.

### 2.4.2. Life satisfaction

Participants completed the five-item Satisfaction with Life scale [47]. This measure includes items like 'In most ways my life is close to ideal' and 'I am satisfied with my life.'

### 2.4.3. Depressive mood (reversed)

Depressive mood was measured using the six-item Adolescent Depressive Mood scale [48]. Participants are asked the extent to which they have been bothered or troubled by tiredness, sleep problems, sadness, hopelessness, nervousness and anxiety. To maintain consistency with the other two well-being measures (i.e. higher scores reflecting greater well-being), depressive mood scale scores were reversed, and thus are referred to here as *absence of depressive mood*.

## 2.5. Dysregulated gaming

### 2.5.1. Video Game Addiction Test

The 14-item Video Game Addiction Test (VAT) [49] was developed as a domain-specific adaptation of the Compulsive Internet Use scale [50]. The VAT assesses loss of control (four items, e.g. 'How often do you continue to use the games, despite your intention to stop?'), conflict (four items, e.g. 'How often do others (e.g. parents or friends) say you should spend less time on games?'), preoccupation/salience (three items), coping/mood modification (two items) and withdrawal symptoms (one item).

### 2.5.2. Clinical Video Game Addiction Test 2.0

The Clinical Video Game Addiction Test 2.0 (CVAT-2) [51] was developed in consultation with clinicians in The Netherlands to conform to the DSM-5 IGD criteria. It was validated on a sample of adolescents and young adults in clinical treatment for dysregulated gaming, exhibiting high sensitivity (91%) in that sample. The CVAT-2 originally takes the form of yes/no checklist. For consistency, this scale was rephrased into statements that are answerable in the five-point format (e.g. 'You regularly play games because you're feeling down' replaces the item 'How often do you game because you are feeling down?'). There exist a 9-item and 11-item version of the CVAT-2, referred to here as CVAT-2 (9-item) and CVAT-2 (11-item). The 9-item version contains items corresponding to each of the nine DSM-5 criteria, while the 11-item version includes two additional items for craving and physical hygiene at the suggestion of therapists involved in the development of this scale. Participants completed the 11-item version, allowing for analysis of both approaches.

### 2.5.3. ICD-11

At the time that this study was conceptualized in 2018, no existing scales covered the criteria outlined for gaming disorder in the ICD-11 [3]. Thus, a four-item scale was drafted based on these criteria, which both read at the time and currently read:

1. Impaired control over gaming (e.g. onset, frequency, intensity, duration, termination, context)
2. Increasing priority given to gaming to the extent that gaming takes precedence over other life interests and daily activities
3. Continuation or escalation of gaming despite the occurrence of negative consequences of sufficient severity to result in significant impairment in personal, family, social, educational, occupational or other important areas of functioning.

A fourth criterion specifies that the above behaviours should be 'evident over a period of at least 12 months.' To incorporate this, one item was generated for each of the three above criteria (e.g. 'You continuously give gaming priority over other hobbies and daily activities'), and a fourth assesses the duration of any problems (Gaming has been problematic for you for over a year).

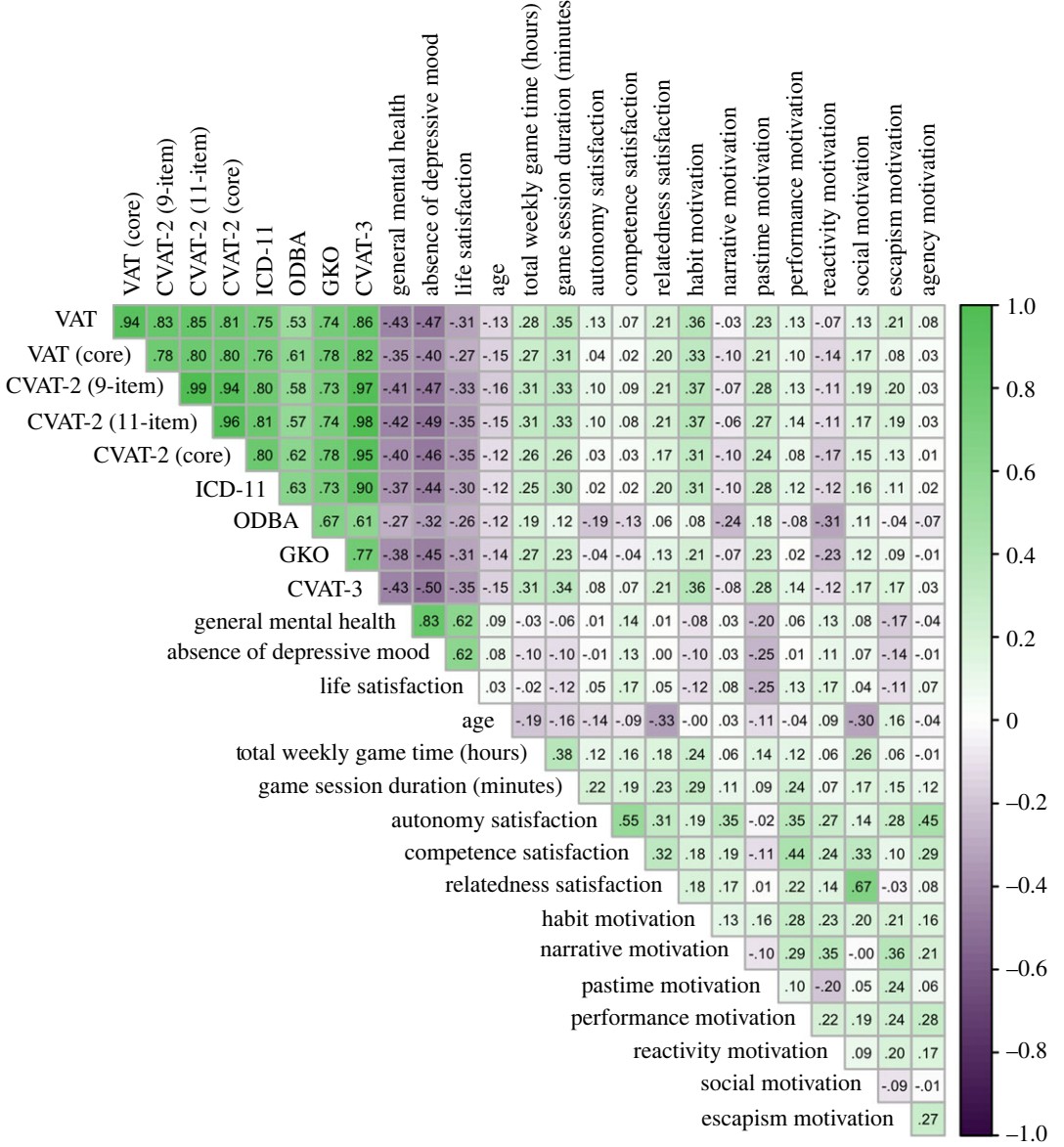

**Figure 1.** Correlation matrix for dysregulated gaming measures, well-being measures and covariates. Darker colours are stronger correlations. Significance labels are omitted for readability; all correlations whose absolute values are greater than $r = 0.16$ are significant at the $p < 0.001$ level.

### 2.5.4. Global Kids Online

Participants completed the five-item Excessive Internet Use sub-scale of the Global Kids Online (GKO) survey [52], which was collaboratively developed for use with 9–17-year olds to understand digital behaviour and health. The Excessive Internet Use sub-scale contains five items assessing functional consequences of dysregulated use of the Internet. The scale was adapted to refer to games specifically, resulting in items like 'I have gone without eating or sleeping because of the time I spent on gaming' and 'my grades have dropped because of the time I spent on gaming.'

### 2.5.5. Open definition of behavioural addiction

Next, the participants completed a four-item scale developed for the purposes of this study based on the open definition of behavioral addiction (ODBA) [53,54],[1] a collaborative effort to form a consensus definition of behavioural addictions across domains. At the time of writing, this definition reads, 'A

---

[1]See also the up-to-date version maintained at https://osf.io/q2vva/.

**Table 2.** Fit indices for each of the nine dysregulated gaming scale operationalizations. CFI, comparative fit index, RMSEA, root mean square error of approximation, SRMR, standardized root mean square residual. Point estimates and confidence intervals for $\omega_h$ are calculated using 1000 bias-corrected and accelerated bootstrap samples, as recommended by [42].

| | $\chi^2$ (d.f.) | CFI | RMSEA [90% CI] | SRMR | McDonald's $\omega_h$ |
|---|---|---|---|---|---|
| VAT | 895.19 (77), $p < 0.001$ | 0.695 | 0.156 [0.145,0.167] | 0.089 | 0.86 [0.84,0.89] |
| VAT (core) | 64.83 (14), $p < 0.001$ | 0.952 | 0.086 [0.058,0.116] | 0.044 | 0.83 [0.78,0.87] |
| CVAT-2 (9-item) | 110.89 (27), $p < 0.001$ | 0.931 | 0.082 [0.063,0.101] | 0.048 | 0.83 [0.80,0.86] |
| CVAT-2 (11-item) | 205.58 (44), $p < 0.001$ | 0.904 | 0.089 [0.074,0.104] | 0.054 | 0.86 [0.83,0.88] |
| CVAT-2 (core) | 62.84 (14), $p < 0.001$ | 0.949 | 0.087 [0.061,0.114] | 0.043 | 0.82 [0.79,0.86] |
| ICD | 3.66 (2), $p = 0.161$ | 0.999 | 0.033 [0.000,0.122] | 0.014 | 0.83 [0.78,0.86] |
| ODBA | 6.64 (2), $p = 0.036$ | 0.989 | 0.068 [0.000,0.147] | 0.026 | 0.63 [0.55,0.71] |
| GKO | 21.51 (5), $p = 0.001$ | 0.989 | 0.075 [0.006,0.135] | 0.025 | 0.85 [0.80,0.88] |
| CVAT-3 | 439.36 (90), $p < 0.001$ | 0.887 | 0.091 [0.080,0.102] | 0.055 | 0.90 [0.88,0.92] |

repeated behaviour leading to *significant harm or distress*. The behaviour is *not reduced by the person* and *persists over a significant period of time*. The harm or distress is of a *functionally impairing* nature' (emphasis added). To the authors' knowledge, no gaming disorder measure has explicitly adopted this definition, and so a four-item measure was drafted on an exploratory basis to assess these criteria (e.g. 'your gaming has been causing problems for you for months, but you are unable to play less.' In contrast with other scales, one item was negatively phrased: 'Gaming does *not* lead to problems in your daily life (such as: problems with school, work, other interests, family, or friends).' The final item assesses one of the ODBA's listed exclusion criteria, and is also reverse-scored: 'The time you spend on gaming is a conscious choice.'

### 2.5.6. Composite scale: DSM-5 (CVAT-2) and ICD-11: CVAT-3

The data for the current study were originally collected to improve the self-test available on the Dutch public health website gameninfo.nl. This website currently makes use of a composite test that covers both the DSM-5 criteria (via the CVAT-2 11-item version) and the ICD-11 items listed above (11 + 4 = 15 items total). The composite scale was chosen for conceptual reasons, namely to yield comprehensive coverage of the criteria in the two most prominent diagnostic manuals. Given that it effectively constitutes an expanded version of the CVAT-2, we refer to this here as the CVAT-3.

## 2.6. Covariates

### 2.6.1. Demographics

Participants reported their age, gender and level of education. The latter was standardized between countries in the dataset to two levels, namely primary education/secondary education versus higher/tertiary education.

### 2.6.2. Game time

Participants provided self-report estimates of the amount of time in hours they spend playing games on an average weekday and average weekend day. These were multiplied (5 × weekday + 2 × weekend) to yield an estimate of participants' total weekly game time.

### 2.6.3. Game session duration

Game session duration was measured with a single item asking participants 'How many minutes do you game, on average, when you play (per game session)?'

### 2.6.4. Gaming motivations

Participants completed an abbreviated version of the Digital Games Motivation scale (DGMS) [55,56]. The DGMS is informed by social cognitive theory [57] and assesses the degree to which players experience eight different identified motives for playing games. The complete DGMS is 43 items, and thus an abbreviated 24-item version is used here, containing three items from each of the DGMS sub-scales, which are:

— *habit*, playing automatically as part of a normal routine
— *moral self-reflection/reactivity*, playing because of expectations of meaning and value from time spent gaming
— *pastime*, playing games to kill time
— *performance*, playing to succeed or excel
— *agency*, playing to experience a sense of volition and play according to one's own preferences
— *narrative*, playing to feel a sense of involvement or believability in the game's story, world and characters
— *social* playing to enact non-competitive behaviour or be respected by other players
— *escapism*, playing to leave the daily routine behind and experience things not possible outside the game world.

### 2.6.5. Need satisfaction in video games

Last, participants completed an adapted version of the Basic Psychological Need Satisfaction and Frustration scale [58]. The adapted version was developed for use in an exercise context with a Dutch sample by [59]. Items were reworded to suit a video games context. The frustration sub-scales are not used, leaving three sub-scales of four items each: autonomy satisfaction (e.g. 'I felt that the game allowed me to do what I wanted to do'), competence satisfaction (e.g. 'I was confident that I was skilled enough to play the game well') and relatedness satisfaction (e.g. 'I felt close and connected to one or more players in-game').

## 2.7. Analytical approach

We adopt specification curve analysis (SCA) [60] as our approach for examining the correlation between the explanatory (gaming disorder severity) and criterion variables (psychological well-being). This type of analysis is well-suited to a dataset like the one at hand in which the presence of multiple similar measures allows for a high degree of analytical flexibility.

The conceptual basis of SCA is to identify all of the theoretically justifiable analysis decisions in the *garden of forking paths* [61] for a given relationship—in this case, dysregulated gaming and well-being—and computing results for all of them. Following Orben & Przybylski's [62] study on self-reported quantity of technology use and well-being, we chose to adopt the simplest possible statistical model, linear regression, which meant that the analysis pathways were determined by (i) how to operationalize well-being, (ii) how to operationalize dysregulated gaming severity, (iii) which covariates to include, and (iv) whether to exclude possible outliers.

Inferences are drawn based on the distribution of the results in the specification curve, which may therefore be interpreted as essentially a single test of the relationship in question, in this case between dysregulated gaming and well-being. As we are not interested in rejecting a null hypothesis based on the results of an individual model, we do not correct for multiple tests. It is nonetheless important to note that a certain proportion of models may be expected to be significant even when there is no true effect (or, alternatively, some models may be non-significant even if there is a true effect, depending on the power of the test). With our sample of 424 participants, we have 90% power to detect a standardized regression coefficient of $\beta = 0.16$ in bivariate linear models (i.e. models with no included covariates).

### 2.7.1. Mental well-being

With regard to the first analytical decision, well-being, we identified all three relevant measures as justifiable specifications, namely life satisfaction (Satisfaction with Life scale), general mental health

(Mental Health Inventory) and (absence of) depressive mood (Adolescent Depressive Mood scale), leading to three specifications.

### 2.7.2. Gaming disorder

For operationalizing gaming disorder, we included all five original measures in the study (VAT, CVAT-2, ICD-11, ODBA and GKO), as well as the combined scale CVAT-3. Both the 9- and 11-item versions of the CVAT-2 were considered valid analysis choices. Given the debates identified above around the inclusion of peripheral versus core criteria, we also identified specifications using only core criteria as justifiable. This resulted in a VAT (core) specification composed of eight items explicitly addressing core criteria as well as a CVAT-2 (core) specification made up of seven items (see tables 5 and 6 in the appendix for exact items). The ICD-11, ODBA and GKO scales each contain five or fewer items, of which only one ODBA item is peripheral, and thus separate core criteria operationalizations were not included for those scales. A core criteria version of the CVAT-3 was deemed to be redundant, as it would consist of the CVAT-2 (core) and all four ICD-11 items, and also was not included. This resulted in nine total specifications for gaming disorder.

### 2.7.3. Covariates

Finally, we identified a number of possible analytical decisions with regard to control variables. To determine which could be justifiably included, we searched the literature for precedent and included all variables that had previously been associated with increased gaming disorder severity. These included (1) age (e.g. [63]), (2) gender (e.g. [64]), (3) educational level (e.g. [65]), (4/5) average amount of time played during a given session and in an average week (e.g. [66]), (6–8) satisfaction of the needs for autonomy, competence and relatedness during video game play (e.g. [37]), and (9–16) the eight gaming motivations (e.g. [30]). Possible model specifications included models with a single covariate, all 16 covariates together, and no covariates, leading to a total of 18 possible specifications for control variables.

### 2.7.4. Outliers

Of the 424 participants, 21 were identified as probable outliers based on high reported weekly game time or long average session duration (three or more standard deviations above the mean, equivalent to 90 weekly hours or 412 min played per session). In addition, one participant reported a total weekly game time of 0 h. However, given that these participants passed the careless response checks and did not exhibit any other signs of malicious reporting (e.g. uniform or patterned responses) and that the focus of dysregulated gaming research is on extreme behaviour, we considered both exclusion or inclusion of the probable outliers as justifiable analysis decisions, and therefore include this as analysis choice within the specification curve. In total, this meant that $3 \times 9 \times 18 \times 2 = 972$ models were calculated.

## 3. Results

We use R v. 4.0.3 [67] and the R packages *dplyr* [68], *ggplot2* [69], *lavaan* [70], *papaja* [71] and *specr* [72] for our analyses and to generate the manuscript. The data, electronic supplementary material, as well as all processing and analysis scripts are available on the Open Science Framework (https://osf.io/h9kmv/); the raw data were originally archived on Dataverse (https://doi.org/10.34894/FMR1WG).

### 3.1. Factor analysis

First, we briefly present data on the psychometric structure of the dysregulated gaming measures both to give context to the results of the specification curve as well as to evaluate the appropriateness of these scales, some of which are novel, for future use. To do so, we run confirmatory factor analyses using robust maximum-likelihood estimation on each of the nine dysregulated gaming scales as well as compute reliability. While we do not adopt absolute cut-off values, fit indices may be interpreted according to the following benchmarks [73]:

— CFI ≥0.95 considered good fit, ≥0.90 considered acceptable
— RMSEA ≤0.06 considered good fit, ≤0.08 considered acceptable
— SRMR ≤0.08 considered good fit

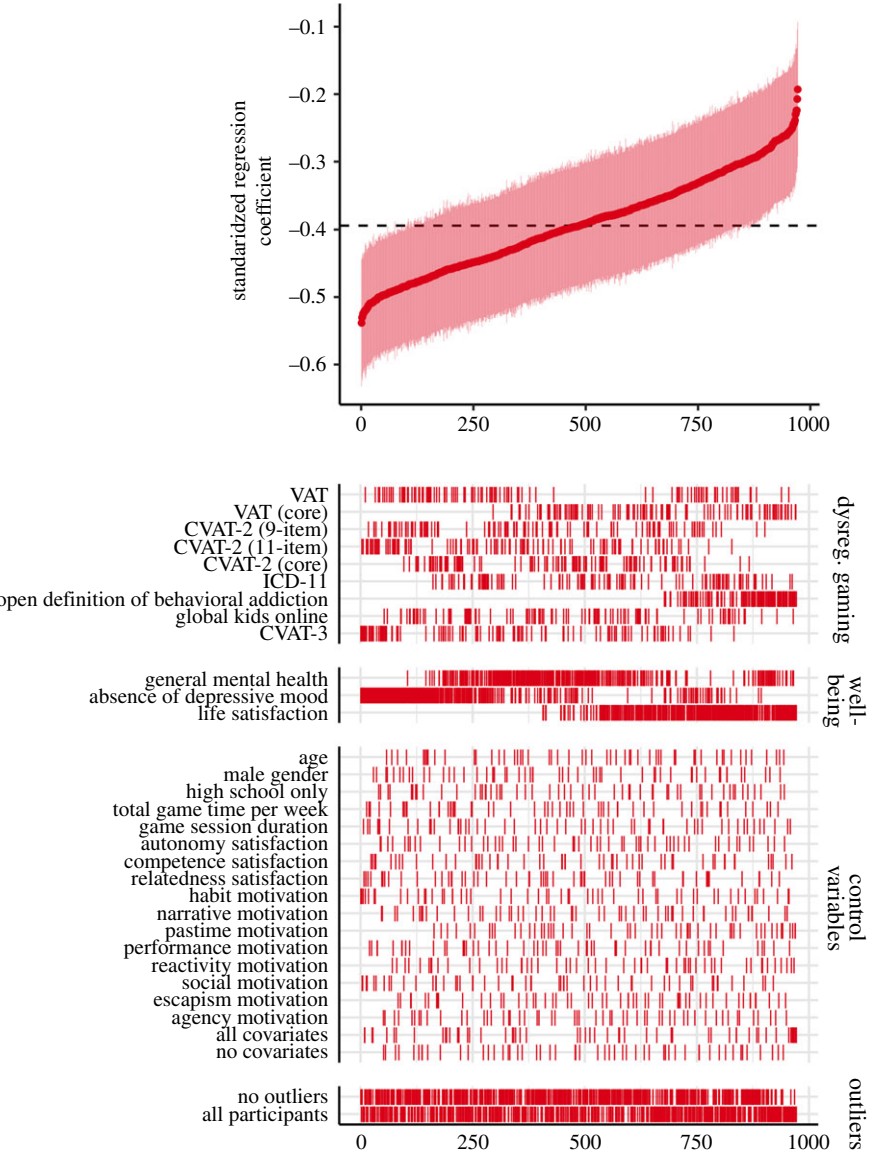

**Figure 2.** Specification curve results comparing the effect size estimates from each of the possible models. In the top portion of the plot, standardized regression coefficients ($\beta$s) for dysregulated gaming as a predictor of well-being are ordered by size. Tightly spaced dark red dots each correspond to the effect size estimate for a single model in which well-being is regressed onto a dysregulated gaming measure and possible covariate(s); light red vertical lines depict the 95% confidence interval of that estimate. The dashed line represents the median effect size of $\beta = -0.40$. In the bottom portion of the plot, each row depicts the distribution of models in which that variable or subset of the data was part of the analysis; each line represents a single model and is aligned vertically with its corresponding effect size estimate in the top part. All effect sizes were calculated with linear regression models and are reported as standardized beta coefficients.

Results of these analyses are summarized in table 2. These suggest the strongest fit for the ICD-11 scale. Fit indices indicated good fit for the ODBA scale as well, but with low reliability, which is influenced by the length of the scale but may also be an indication of an insufficiently narrow construct [74]. Results for both the ICD-11 and ODBA scales should be interpreted cautiously, as with only four items per scale the models have only two degrees of freedom, leading to wide confidence intervals for RMSEA. However, evidence suggests that certain fit indices like RMSEA over-reject correctly specified models with few degrees of freedom [75], and thus we have a reasonable degree of confidence in the performance of these scales.

Fit is somewhat worse but generally acceptable for both versions of the CVAT-2 as well as the CVAT-3, and the results suggest poorest fit for the VAT. The poor fit for the VAT compared with its initial validation study may be in part a function of correlated residuals used in the original factor model;

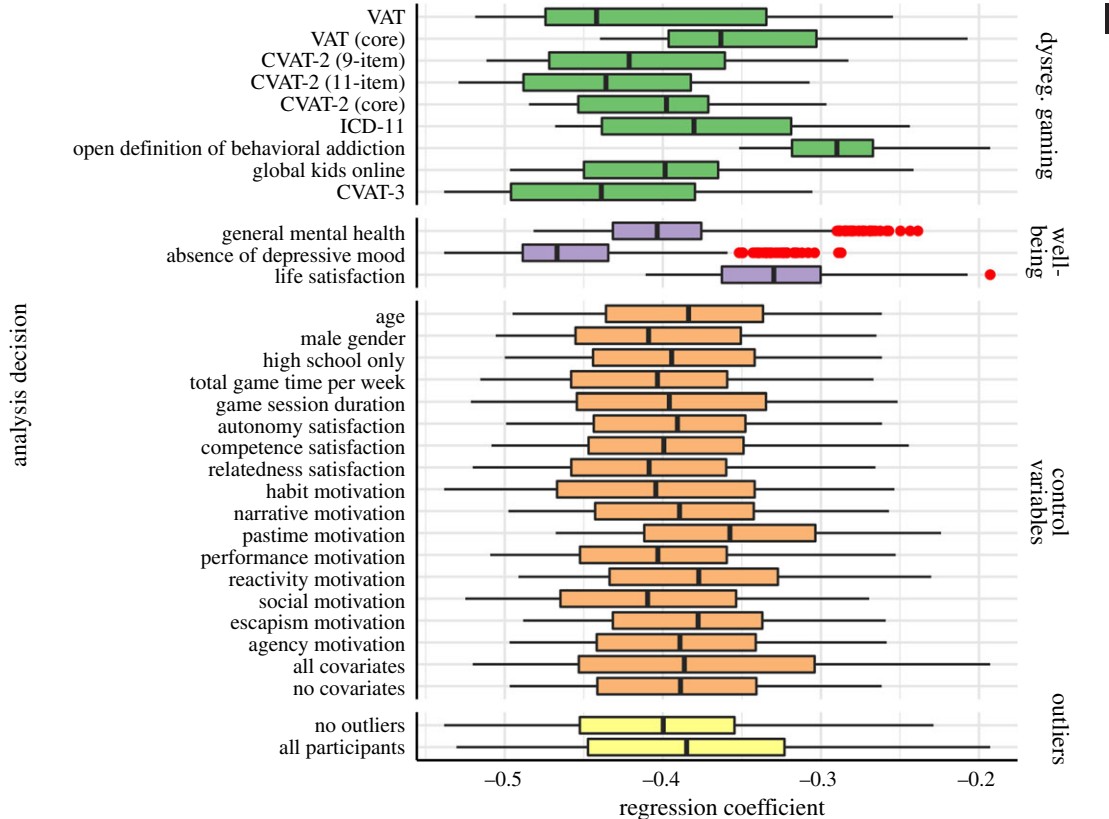

**Figure 3.** Box plots corresponding to the bottom half of figure 2. Each box plot shows the distribution of regression coefficients for the primary relationship between dysregulated gaming and well-being for all SCA models adopting that analytical decision. Vertical lines represent the median effect size for that analysis choice, while red dots indicate outliers whose effect sizes in the SCA were at least $\pm 1.58 \times IQR/\sqrt{n}$.

because this study uses mean scores instead of latent factor scores, this is not accounted for here. Fit is drastically improved, however, in the core version of the VAT relative to the full version ($\Delta$CFI = 0.257; $\Delta$RMSEA = −0.070). The same is true to a lesser extent for the CVAT-2 (core) compared with the full version ($\Delta$CFI = 0.045; $\Delta$RMSEA = −0.002). Overall, factor analysis results are mixed, with some indices like SRMR evidencing good fit for most models, but others like RMSEA suggesting poorer fit. Results from the VAT (full version) should be interpreted with the most caution, while the remainder appear to meet minimum standards of measurement validity.

## 3.2. Specification curve

### 3.2.1. Dysregulated gaming measures

All of the 972 models under investigation yielded significant negative relationships between dysregulated gaming and mental well-being (figure 2). The median dysregulated gaming coefficient across all models was $\beta = -0.39$, with the median model accounting for 15.9% of the variance (multiple $R^2 = 0.159$). The model with the smallest coefficient regressed ODBA scores onto life satisfaction with all covariates, resulting in an effect size of $\beta = -0.19$ (95% CI [−0.29, −0.09]; multiple $R^2 = 0.188$). The model with the largest coefficient estimate, on the other hand, uses the CVAT-3 as the $x$ variable and depressive mood as the $y$ variable, with habit motivation as a control ($\beta = -0.54$; 95% CI [−0.63, −0.44]; multiple $R^2 = 0.246$).

We find substantial overlap in the distribution of estimates across the various dysregulated gaming measures (figure 3). On average, the VAT, CVAT-2 (11-item) and CVAT-3 predicted the highest amount of variance in well-being among dysregulated gaming specifications (i.e. yielded the largest effect sizes), but distributions for these largely coincide with the remaining measures. This was also the case for the comparison of core and peripheral criteria; data show minimal differences between the core criteria versions of the VAT and CVAT-2 (11-item) and their respective full versions, with the core

**Table 3.** Summary of linear regression coefficients for each predictor of well-being (dysregulated gaming measures, demographics and covariates) across all models in which they appear.

| variable | median $\beta$ [95% CI] | min $\beta$ | max $\beta$ | median $\Delta R^2$ |
|---|---|---|---|---|
| **dysregulated gaming** | | | | |
| VAT | −0.44 [−0.53, −0.35] | −0.52 | −0.25 | 0.184 |
| VAT (core) | −0.36 [−0.46, −0.27] | −0.44 | −0.21 | 0.122 |
| CVAT-2 (9-item) | −0.42 [−0.51, −0.33] | −0.51 | −0.28 | 0.164 |
| CVAT-2 (11-item) | −0.44 [−0.53, −0.35] | −0.53 | −0.31 | 0.177 |
| CVAT-2 (core) | −0.40 [−0.49, −0.31] | −0.48 | −0.30 | 0.151 |
| ICD-11 | −0.38 [−0.47, −0.29] | −0.47 | −0.24 | 0.137 |
| open definition of behavioural addiction | −0.29 [−0.38, −0.20] | −0.35 | −0.19 | 0.079 |
| Global Kids Online | −0.40 [−0.49, −0.31] | −0.50 | -0.24 | 0.147 |
| CVAT-3 | −0.44 [−0.53, −0.35] | -0.54 | −0.31 | 0.180 |
| **demographics** | | | | |
| age | 0.04 [−0.05, 0.13] | −0.02 | 0.09 | 0.001 |
| †male gender | 0.43 [0.22, 0.63] | 0.05 | 0.58 | 0.030 |
| †high school only | 0.15 [−0.04, 0.33] | −0.09 | 0.27 | 0.005 |
| total game time per week | 0.05 [−0.06, 0.17] | −0.09 | 0.16 | 0.001 |
| game session duration | 0.01 [−0.10, 0.11] | −0.13 | 0.11 | 0.000 |
| **covariates** | | | | |
| autonomy satisfaction | 0.00 [−0.10, 0.11] | −0.10 | 0.10 | 0.000 |
| competence satisfaction | 0.12 [0.02, 0.22] | 0.06 | 0.22 | 0.009 |
| relatedness satisfaction | 0.07 [−0.03, 0.18] | −0.08 | 0.15 | 0.004 |
| habit motivation | 0.00 [−0.10, 0.10] | −0.14 | 0.11 | 0.000 |
| narrative motivation | −0.00 [−0.10, 0.09] | −0.07 | 0.07 | 0.000 |
| pastime motivation | −0.13 [−0.22, −0.03] | −0.21 | −0.05 | 0.014 |
| performance motivation | 0.10 [0.00, 0.19] | −0.01 | 0.22 | 0.008 |
| reactivity motivation | 0.07 [−0.03, 0.16] | −0.02 | 0.17 | 0.004 |
| social motivation | 0.11 [0.00, 0.21] | −0.04 | 0.17 | 0.007 |
| escapism motivation | −0.08 [−0.18, 0.01] | −0.18 | −0.01 | 0.005 |
| agency motivation | 0.01 [−0.09, 0.10] | −0.05 | 0.10 | 0.000 |

†Unstandardized estimate. Medians and 95% confidence intervals reflect the results of either the individual regression model containing the central-most estimate across all models that include the variable, or, where the number of models with a given predictor is even, the average of the two most central estimates and their standard errors. Estimates for non-binary gender and primary school education level are not included here due to sparsity of data. Median $\Delta R^2$ (semipartial $R^2$) can be interpreted as the squared correlation between that predictor and the outcome variable after accounting for collinearity between predictors.

criteria versions of each producing on average slightly less negative estimates than their core/peripheral counterparts, but with overlapping confidence intervals (table 3).

One scale proved to be an exception; visual inspection of the SCA plot as well as investigation of the effect sizes in table 3 revealed that the ODBA scale was the dysregulated gaming measure most weakly associated with well-being. The range in median point estimates for the eight other dysregulated gaming measures spanned from −0.44 to −0.36 ($\Delta\beta = 0.08$). The difference between the median point estimate for the ODBA scale and its closest other scale, the ICD-11, was 0.09.

## 3.3. Well-being

The results show that distributions of estimates differ between well-being measures (figure 3), with depressive mood exhibiting the largest coefficients on average (median $\beta = -0.47$,

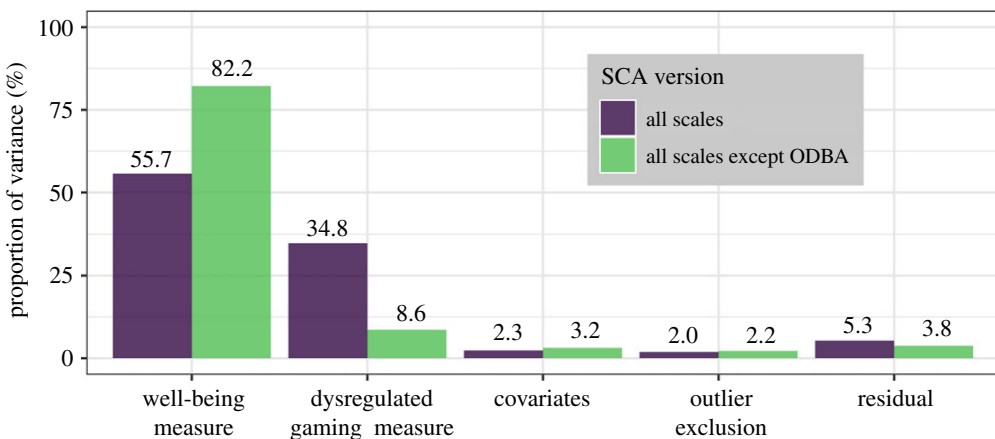

**Figure 4.** Decomposed variance of the specification curve analyses. Variance here refers to the distribution of regression coefficients for dysregulated gaming as a predictor of well-being across the 972 SCA models (i.e. variance leading to the observed effect size range of −0.19 to −0.54). Bars indicate the percentage of this variance that is explained by analytical decisions involving each component of the models (summing to 100%). Purple bars are results from the first SCA with all nine possible operationalizations of dysregulated gaming, while green bars reflect results of the same analysis conducted after removing the ODBA as a specification after data indicated that this measure was anomalous.

95% CI [−0.55, −0.38]; min $\beta$ =−0.54, [−0.63, −0.44]; max $\beta$ =−0.29, [−0.38, −0.19]). General mental health showed a slightly weaker relationship (median $\beta$ =−0.40, 95% CI [−0.50, −0.31]; min =−0.48 [−0.58, −0.38]; max =−0.24 [−0.34, −0.14]), with life satisfaction having on average the weakest relationship (median $\beta$ =−0.33, 95% CI [−0.42, −0.23]; min =−0.41 [−0.51,−0.32]; max =−0.19 [−0.29,−0.09]).

## 3.4. Decomposed variance

To better understand the relative importance of analytical decisions concerning each component of the models (i.e. dependent variable, independent variable and controls), we conducted a decomposed variance analysis. This method calculates intraclass correlation coefficients using a multilevel model with no fixed effects, but with random effects representing the different analytical choices (a so-called null model). This quantifies the contribution of each class of analysis choice to the total variance in effect sizes observed across the specification curve models [72], see also [76] for details on intraclass coefficients.

Results of the decomposed variance analysis show that the choice of dysregulated gaming measure accounts for 34.8% of the variance in coefficients (figure 4). The choice of well-being outcome, on the other hand, was more important, accounting for 55.7% of the variance. Analytical decisions regarding the covariates and the inclusion of outliers were the least impactful, at only 2.3% and 2.0% of the variance, respectively.

As noted above, results indicated that the ODBA scale was unique among the nine dysregulated gaming specifications. Thus, for comparison, we repeated the same SCA but omitted the ODBA, and conducted the decomposed variance analysis again. With the ODBA omitted, analytical decisions involving the remaining eight dysregulated gaming specifications accounted for just 8.6% of the variance, with well-being now accounting for 82.2%, showing that the ODBA adds substantial additional variance to the specification curve when included and confirming its divergent behaviour.

## 3.5. Covariates

Table 3 shows that the majority of covariates account for very little unique variance. Of the control variables, being male was most strongly linked with well-being (median *unstandardized* coefficient $b$ = 0.43, 95% CI [0.22, 0.63]). Being male was more strongly associated with general mental health and absence of depressive mood (median $b$s = 0.46 and 0.45, respectively) than with life satisfaction (median $b$ = 0.14).

The eight gaming motivations sub-scales and three need satisfaction in games sub-scales had small and inconsistent effects (median $\beta$s ≤ 0.13; median $\Delta R^2$s ≤ 0.014). Total weekly game time, while

**Table 4.** Effect sizes from specification curve using single-item indicators of dysregulated gaming severity as explanatory variables. Items are ordered from largest to smallest median $\beta$.

| variable | item label | median $\beta$ [95% CI] | min $\beta$ | Max $\beta$ | median $\Delta R^2$ |
|---|---|---|---|---|---|
| mood modification/escapism | CVAT-2#5 | −0.46 [− 0.55, −0.37] | −0.52 | −0.28 | 0.201 |
| problems (health) | CVAT-2#11 | −0.36 [− 0.45, −0.27] | −0.40 | −0.30 | 0.130 |
| loss of interests | CVAT-2#6 | −0.35 [− 0.44, −0.26] | −0.40 | −0.28 | 0.122 |
| continuation despite problems (ICD) | ICD-11#3 | −0.34 [− 0.43, −0.25] | −0.40 | −0.21 | 0.112 |
| problems persist > 1 year (ICD) | ICD-11#4 | −0.32 [− 0.41, −0.23] | −0.41 | −0.21 | 0.098 |
| loss of control (ICD) | ICD-11#1 | −0.30 [− 0.39, −0.20] | −0.35 | −0.11 | 0.084 |
| loss of control | CVAT-2#3 | −0.29 [− 0.39, −0.20] | −0.32 | −0.10 | 0.084 |
| lying/deception | CVAT-2#7 | −0.28 [− 0.38, −0.19] | −0.32 | −0.13 | 0.079 |
| withdrawal | CVAT-2#2 | −0.28 [− 0.38, −0.19] | −0.35 | −0.12 | 0.075 |
| loss of interests (ICD) | ICD-11#2 | −0.28 [− 0.37, −0.18] | −0.37 | −0.19 | 0.073 |
| craving | CVAT-2#10 | −0.27 [− 0.36, −0.17] | −0.34 | −0.19 | 0.069 |
| problems (work/social) | CVAT-2#9 | −0.23 [− 0.32, −0.13] | −0.32 | −0.16 | 0.049 |
| preoccupation | CVAT-2#4 | −0.21 [− 0.31, −0.11] | −0.25 | −0.04 | 0.040 |
| continuation despite problems | CVAT-2#8 | −0.21 [− 0.30, −0.11] | −0.29 | −0.16 | 0.042 |
| tolerance | CVAT-2#1 | −0.17 [− 0.27, −0.07] | −0.24 | −0.11 | 0.028 |

exhibiting moderate zero-order correlations with dysregulated gaming severity and a weak zero-order correlation with absence of depressive mood ($r = −0.097$, 95% CI [−0.190, −0.002], $p = 0.046$), was not correlated with either life satisfaction or general mental health, and was a negligible independent predictor of well-being in the regression models (median $\Delta R^2 = 0.001$).

## 3.6. Item-level analyses

To better understand the finding that the VAT (core) and CVAT-2 (core) specifications diverged minimally from their core and peripheral counterparts, we conducted a final exploratory SCA, this time using individual scale items as the possible predictors of well-being. We chose the CVAT-2 (11-item) and the ICD-11 scales for this to fully capture the criteria included in the DSM-5 and ICD-11.

Results show that all individual items are significantly associated with decreased well-being in the majority of models, but that the strength of the relationship varies widely between criteria (table 4).

Despite being a peripheral criterion, the largest single-item predictor of decreased well-being was the CVAT-2's mood modification/escapism item, whose median effect size was $\beta = −0.46$ (95% CI [−0.55, −0.37], $\Delta R^2 = 0.201$). The items with the weakest association with well-being, on the other hand, were tolerance (median $\beta = −0.17$ 95% CI [−0.27, −0.07], $\Delta R^2 = 0.028$), continuation despite problems ('Did you play games even though you knew this was causing problems with your family, friends, at work, or at school?') ($\beta = −0.21$ 95% CI [−0.30, −0.11], $\Delta R^2 = 0.042$) and preoccupation ($\beta = −0.21$ 95% CI [−0.31, −0.11], $\Delta R^2 = 0.040$).

# 4. Discussion

## 4.1. Dysregulated gaming and well-being

The results of the specification curve analysis show a consistent negative relationship between dysregulated gaming severity and mental well-being. Regardless of how one operationalizes

dysregulated gaming, well-being, outliers and covariates, the effect remains statistically significant. Relative to the minimal existing data on the role of analytical flexibility in studies on digital media, the distribution of effect sizes in our study forms a comparably narrow range. For example, [62] find in one of their three datasets that analytical flexibility in the relationship between technology use and well-being can lead to effect size estimates ranging from approximately $\beta = -0.90$ to $\beta = 0.05$, a total range of 0.95, and that approximately 20% of models yield non-significant effects despite a large sample size and high precision. In our study, we find a total range in effect sizes of 0.35 with all models yielding significant results, providing evidence that dysregulated gaming is more robustly related to well-being and is less subject to the influence of analytical flexibility than research on screen time, a topic known to have particular measurement challenges (e.g. [77]).

The data indicate, however, that dysregulated gaming is not uniformly associated with different aspects of well-being (with choice of well-being measure accounting for 82.2% of the variance between models, figure 4). Dysregulated gaming was most strongly associated with greater depressive mood, followed by decreased general mental health, and least strongly with lower life satisfaction. This suggests that there may be variation in the relationship between dysregulated gaming and well-being both at the individual level and across constructs at the population level. While it remains unclear which aspects of well-being are most likely to be linked with dysregulated use and the factors that may underlie this, our findings mirror a recent review article which finds evidence that 'digital technology use has stronger effects on short-term markers of hedonic well-being (e.g. negative affect) than long-term measures of eudaimonic well-being (e.g. life satisfaction)' [78, p. 135].

Despite some differences in dysregulated gaming's relationship with general mental health, depressive mood and life satisfaction, all effects were relatively large in size, ranging from $\beta = -0.54$ to $\beta = -0.19$ with a median of −0.39. Effect sizes are difficult to directly interpret, and may be best understood in relation to other, related effects. A full comparison of dysregulated gaming, technology use and well-being measures is outside the scope of this study, and is made more difficult by the fact that a large portion of the published literature has dichotomized participants into 'non-IGD' and 'probable IGD' groups; these are analysed with logistic regressions that cannot be straightforwardly compared with the linear regression models used here. However, a few examples may be illustrative.

The results presented here diverge somewhat from an early meta-analysis of problematic gaming and mental health, which found only a small correlation between problematic gaming and decreased mental health of $R^2 = 0.036$ (95% CI [0.017, 0.058]), and an $R^2$ of 0.068 (95% CI [0.027, 0.130]) among studies that used core criteria only [79]. Similarly, we find a larger association than a more recent meta-analysis [80], which reports a correlation between gaming disorder and psychological well-being of $r = -0.146$ ($R^2 = -0.021$) across 66 studies. The effect sizes in the current work broadly align with some other non-meta-analytic studies; however, one study found that gaming disorder accounted for 20% of the variance in loneliness and 22% of the variance in depression, slightly higher than both the zero-order correlations and partial $R^2$ values reported here [40].

It is critical to note that the relationship between dysregulated gaming severity and well-being described here cannot be interpreted as causal. While we find relatively strong associations in this study, these can be contrasted with longitudinal studies that can offer causal evidence, the results of which have not been consistent. Scharkow et al. [14], for example, found little to no evidence of an effect of problematic gaming on life satisfaction in a 2-year longitudinal study. In a 2-year study of problematic gaming in Singapore, on the other hand, dysregulated gaming severity at time 1 was associated with both depression ($\beta = 0.37$) and anxiety ($\beta = 0.29$) 2 years later; the authors do not, however, report causal effects in the opposite direction (i.e. depression on subsequent dysregulated gaming) [8]. Another study found that a gaming disorder classification at time 1 predicted emotional distress and hyperactivity one year later; these effects were smaller, however, than that of hyperactivity on subsequent gaming disorder [81]. These comparisons are limited in scope, however, as longitudinal data and gold-standard causal inference remain severely lacking in the field [82].

In sum, we find that one can expect certain well-being outcomes—possibly those related to hedonic well-being—to be more strongly linked with dysregulated gaming than others. Well-being outcomes therefore cannot and should not be straightforwardly substituted for one another. Further investigations using additional aspects well-being (e.g. anxiety, self-esteem, social health) and other functional outcomes (e.g. work/academic performance, physical health) may help to identify the aspects of a person's life that are most likely to be negatively linked to dysregulated gaming, and whether this may be moderated by personality factors. This may in turn help with identification of dysregulation, the development of adaptive coping strategies, and the provision of effective clinical treatment where necessary.

## 4.2. Dysregulated gaming measure selection and validity

While we find evidence of meaningful differences in effect size depending on the selected well-being measure, the choice of dysregulated gaming measures introduced minimal variance. With the exception of the scale based on the open definition of behavioural addiction (ODBA), results showed that the other eight scale specifications converge upon similar associations with decreased well-being, that this is true even for shorter scales, and that it remains true even for scales with weaker model fit. Thus, we do not conclude that any of the scales used here are more predictive of decreased well-being than others.

The reasons underlying the anomalous behaviour of the ODBA are not clear, but appear to be driven in part by one item which stands out conceptually among all of the scale items, namely 'the time you spend on games is a conscious choice.' This assesses the role of self-reflection in gaming and is not typically found among dysregulated gaming measures, and is indeed the item that loads most poorly onto its hypothesized latent factor ($\lambda = 0.28$). More work is necessary to operationalize the definition put forth in the ODBA [54] and to understand the relationship between non-reflective or low self-awareness game use and well-being outcomes.

Despite repeated calls for consensus and unity in the assessment of dysregulated gaming, measurement tools continue to proliferate [17]. The evidence presented here indicates that measures may be largely convergent in their relationship with the psychosocial and functional outcomes in which we are ultimately interested (i.e. in terms of concurrent criterion validity). We suggest that time and resources may be better spent on topics like harmonization of measurement, comorbidity, the ontology and temporal stability of dysregulation, and treatment efficacy, rather than continued validation of new or existing scales used for cross-sectional studies and non-clinical populations.

## 4.3. Core versus peripheral criteria

Contrary to expectations, the two scales assessing only core criteria differed only slightly and non-significantly in their association with well-being from the scales assessing both core and peripheral criteria. This finding diverges somewhat from previous results (e.g. [79]), showing that scales including peripheral criteria as defined here may still be meaningfully associated with decreased well-being. However, the core and peripheral scales did not account for more variance than the core criteria scales alone. We interpret these results to indicate that although the inclusion of peripheral criteria does not add significant amounts of noise such that the relationship between dysregulated gaming and well-being is obscured, neither does it add meaningfully to our ability to predict the negative outcomes investigated here, and may simply add to participant burden.

To better understand this finding, we investigated the relationships between individual criteria and well-being. Some peripheral items followed the expected pattern; tolerance and preoccupation, which have been criticized in previous work [24,25,83], were found to account for minimal variance in well-being. However, the results for the CVAT-2's mood modification/escapism item were contrary to expectations and showed that mood modification, despite being a peripheral criterion, was the single item most strongly associated with decreased well-being.

One possible explanation for this is that the mood modification item may not only be assessing the extent to which people play games to escape negative moods or the difficulties of day-to-day life, but rather the *presence* of negative mood and difficulties in day-to-day life in the first place. When looking at dysregulated gaming and well-being or functional impairment, this would therefore mean that a portion of the outcome is included as a predictor. At the moment, this is a speculative explanation and requires nuanced (and ideally within-person) analyses showing an association between escapism and negative affect after controlling for dysregulated gaming. Were this confound to be confirmed, however, this would have significant implications for both the DSM-5 criteria as well as for the validity of many existing measurement tools. Prominent scales like the IGDS-SF9 [84] and IGD-9 [85] include items like 'Do you play in order to temporarily escape or relieve a negative mood (e.g. helplessness, guilt, anxiety)?' and these items may contribute to inflated prevalence estimates of gaming disorder.

Given the wide range of biological, psychological and ecological factors that can contribute to the development of dysregulated behaviour, some previous research on problematic phone use and problem gambling has adopted a 'pathways' approach, which describes different subgroups for whom behavioural patterns may be similar, but with distinct underlying developmental process having led to the dysregulated behaviour [86,87]. Blaszczynski & Nower [86] outline a distinction between

behaviourally conditioned problem gamblers and emotionally vulnerable problem gamblers for whom gambling 'is viewed as a means of producing emotional escape through the effect of dissociation on mood alteration and narrowed attention' (p. 493), with the latter group also potentially influenced by neurological and neurochemical factors that increase their susceptibility to dysregulated behaviour. In the context of games, it is possible that using games as a coping mechanism for problems in one's life is a separable phenomenon from both non-problematic high game engagement and less-comorbid dysregulated gaming. Lee *et al.* [88] advocate for this type of distinction as well; their model of gaming disorder heterogeneity identifies impulsive/aggressive, emotionally vulnerable, socially conditioned, and not otherwise specified as possible subtypes of the condition.

We note although we do not find evidence between core and peripheral scales in terms of (concurrent) criterion validity, our results do support a psychometric distinction between interference (core) and engagement (peripheral) criteria. Model fit for the VAT (core) and the CVAT-2 (core) are greatly improved relative to the full versions, indicating that the full versions of those scales are not unidimensional. We advise researchers to be cautious if using the VAT or CVAT-2 for cross-sectional research (i.e. where scores are assumed to represent a single latent construct) and either conduct their own factor analyses before interpreting results, or to understand them as comprising a formative rather than reflective construct [89]. The remaining operationalizations, including the core versions of the VAT and CVAT-2, ICD-11, GKO, and to a slightly lesser extent CVAT-3, appear to be justifiable candidates for use in survey-based dysregulated gaming research.

We argue that criteria for dysregulated gaming should be meaningfully associated with functional and psychosocial well-being outcomes and as parsimonious as possible. Our results suggest that a subset of peripheral criteria, notably tolerance, may have small or practically insignificant associations with well-being, and that escapism's large association with well-being may be the result of a confound. Future work in this area will benefit from comparisons of core and peripheral criteria scales using other outcome measures (e.g. academic/career performance, physical health, relationship quality) as well as more in-depth investigations of the criterion validity of individual items.

## 4.4. Role of covariates

Overall, the inclusion or exclusion of covariates had a negligible impact on our results. The covariate with the largest impact was gender; being male was associated with higher general mental health and less depressive mood symptoms, but was only weakly related to life satisfaction in our sample. This is consistent with previous research finding higher rates of depression in women during both adolescence and adulthood [90,91].

Turning to *motivational styles*, which have received significant research attention as possible predictors of dysregulated gaming, we find that three motivations—habit, escapism and pastime—were zero-order correlated with dysregulated gaming measures, and that two of these (escapism and pastime) were correlated with well-being as well, but in the regression models independently predicted very little of the variance in well-being. This is in line with previous evidence (e.g. [27]), and shows that these motivations may help us to understand the underlying causes of dysregulated gaming. However, similar to mood modification items in dysregulated gaming measures (see above), escapism motivation items may in part assess the *presence* of negative mood and difficulties from which to escape. We caution against treating motivational profiles as risk factors, given the lack of clarity regarding causality, but encourage further work to determine why some players with escapist motivations may report lower well-being, while others do not.

We find that the presence of *need satisfaction in games* is a negligible predictor of well-being, and that competence satisfaction in games is even weakly positively associated with well-being on average. This suggests that need satisfaction in video games is not a meaningful risk factor for dysregulated gaming by itself. We note, however, that SDT contrasts between need satisfaction and need frustration [92], and emerging evidence supports a relationship between need frustration in daily life beyond video games (i.e. a persistent sense of lacking control in one's life, ineffectiveness in one's activities, and/or social disconnectedness) and dysregulation [36,37]. While we did not account for daily or in-game need frustration in the current paper, this is a promising area for future work, and may help us understand both the interplay of environmental factors and behaviour, as well as the developmental course of dysregulation.

Finally, we find no evidence that *weekly game time* is meaningfully associated with well-being. Results show weak and non-significant zero-order correlations between dysregulated gaming symptom endorsement and well-being, and across 108 multiple regression models that included weekly game

time as a covariate, none of its coefficients were statistically significant. In other words, after accounting for dysregulated gaming, the relationship between self-reported quantity of game use on well-being ranges from slightly positive to slightly negative, with the median regression model showing a minute positive relationship. These results are comparable and underscore those of Orben & Przbylski [62], whose specification curve analysis of technology use and well-being in adolescents found that the median effect sizes from three large-scale datasets were negligible in size ($\beta$ s = $-0.04$, $-0.01$, $-0.03$; partial $R^2$s = $-0.001$, $< 0.001$, $0.004$). In the context of games specifically, other work has also found small and non-significant effects of self-reported gameplay frequency on psychosocial health [93].

In other words, weekly playtime is weakly correlated with dysregulated gaming severity, and dysregulated gaming severity is moderately correlated with well-being outcomes, but the absence of a clear relationship between weekly playtime and well-being directly suggests that other factors besides quantity of use are more important for understanding problematic patterns of play. Thus, we argue that research attention should be focused on the contextual, motivational and personality factors that may differentiate game use that positively affects players' lives to the rarer cases in which game use is associated with negative consequences, rather than amount of playtime.

## 4.5. Limitations

As with the majority of the studies in the field, this study is limited by overall low rates of dysregulated gaming symptom endorsement. Response patterns and associations in non-clinical samples are likely to differ from clinical populations, and the same is true of samples of people who play games (as gathered here) versus the population at large. The results of this study should also not be assumed to generalize to more severe dysregulation, different well-being constructs, or other cultures.

This study is also limited by the measures included, with some having been newly developed for this study (ICD-11, ODBA, GKO and CVAT-3), one exhibiting mediocre model fit here despite previous evidence of validity (VAT), and others adapted for use outside of their original clinical context (CVAT-2). The choice of included measures was influenced by a few factors. First, survey design decisions were driven by the primary goal of the data collection, namely to develop and validate the CVAT-3 for use as a self-test for Dutch and Flemish speakers, as opposed to an explicit effort to compare the most commonly used gaming disorder scales. Second, Dutch language scales were used, which are neither as widespread nor extensively validated as English scales. Finally, the high participant burden of the survey led to the necessity of limiting the length of any newly developed measures. Results for the measures with weaker model fit should be interpreted cautiously, and further work will be necessary to justify their inclusion in future studies.

The survey design is a further limitation; a fixed order was used to prevent repetitive measures being placed together and to avoid long series of negatively oriented measures. Manual inspection of the data showed minimal signs of straight-lining; where this did occur, these were typically the same participants who failed one or both attention checks, justifying the decision to exclude these participants. We did not observe any systematic change in mean scores or decreasing within-scale variance with measures later in the survey. Nonetheless, (conceptual) replications of the current study and further work on the possibility of order effects would both be valuable.

Finally, the majority of the measures used here do not have empirically grounded diagnostic thresholds. Therefore, it was not possible to assess the relationship between well-being and a gaming disorder diagnosis or classification according to each scale; instead, we focused on severity as continuous variable, i.e. the mean score of gaming disorder measures. While diagnosis based on scale scores are controversial and diverge from clinical appraisals [94], the DSM-5 explicitly states a threshold of five out of nine endorsed criteria. Future work would be well-served to compare the relationship between well-being, clinician-rated dysregulated gaming, and frequently used measures like the IGDS-SF9 [84] that specify a proposed diagnostic cut-off point.

## 5. Conclusion

In this study, we used specification curve analysis to investigate the strength of the relationship between dysregulated gaming and well-being. We find that analytical decisions can result in effect size estimates ranging from $-0.54$ to $-0.19$, but that the direction and significance of the relationship is consistent across all justifiable models. We contextualize this as a comparatively narrow range, and find that while one dysregulated gaming measure was anomalous, only a small portion of the variance in these effect

sizes was attributable to the difference between the remaining eight dysregulated gaming measures. Rather, differences were primarily a result of choice of well-being measure, with life satisfaction most weakly associated with dysregulated play and depressive mood most strongly. Our results indicate that peripheral criteria, while not adding significant amounts of noise to the association between dysregulated gaming and well-being, also do not strengthen this relationship. Mood modification items and related escapism motivations may assess the presence of problems or negative moods in a person's life, and not simply the use of games to cope with problems or moods. Weekly playtime was not meaningfully associated with well-being. We argue for a wider comparison of well-being outcomes, a focus on identifying the most parsimonious list of meaningful gaming disorder criteria, and harmonization of measures that may ultimately differ little from each other in survey-based research.

Data accessibility. All data, code and materials associated with this project are available on the Open Science Framework http://doi.org/10.17605/osf.io/H9KMV.

Authors' contributions. A.J.V.R. designed the initial study and collected the data. N.B. and A.J.V.R. conducted data analysis. N.B. wrote the first draft of the manuscript. Both authors contributed equally to revision and approve the final manuscript.

Competing interests. We declare we have no competing interests.

Funding. The first author was supported by the EPSRC Centre for Doctoral Training in Intelligent Games & Games Intelligence (IGGI) (EP/S022325/1). The second author received funding from the ZonMw project no. 531004006, Preventieprogramma 5.

# Appendix A

List of item wordings for each unique scale in the study. Endorsement percentages refer to the proportion of respondents that answered either 'agree' or 'strongly agree' to each item ('disagree' or 'strongly disagree' for reversed scored items). Endorsement percentages are provided for context only; dichotomizing responses on an ordinal scale is a controversial practice (see tables 5–9).

**Table 5.** Items wordings for the Video Game Addiction Test [49]. The canonical version of the VAT is in Dutch. English translations provided are not the result of a formal translation process; these are for reference only and may diverge slightly from the English translations in the original paper (which are also provided for reference and did not undergo a formal translation process).

| label in data | construct | English wording | Dutch wording | core or peripheral | per cent endorsed |
|---|---|---|---|---|---|
| vat1 | loss of control | you regularly find it difficult to stop gaming | Je vindt het regelmatig moeilijk om met gamen te stoppen | peripheral | 14.6 |
| vat2 | loss of control (failed attempts to stop) | you often continue to play games, despite your intention to stop | Je gaat vaak langer door met gamen, terwijl je je had voorgenomen om te stoppen | core | 30.9 |
| vat3 | conflict (interpersonal) | others (e.g. parents or friends) repeatedly say you should spend less time on games | Anderen (bijv. ouders of vrienden) zeggen steeds dat je minder zou moeten gamen | core | 17.2 |
| vat4 | behavioural salience (preference) | you often prefer to game instead of spending time with others | Je gaat vaak liever gamen dan dat je je tijd met anderen doorbrengt | peripheral | 21.2 |

(Continued.)

| label in data | construct | English wording | Dutch wording | core or peripheral | per cent endorsed |
|---|---|---|---|---|---|
| vat5 | problems (health) | you regularly do not get enough sleep because of gaming | Je komt regelmatig slaaptekort door het gamen | core | 11.6 |
| vat6 | preoccupation (cognitive salience) | you often think about gaming, even when you cannot play | Je bent vaak in gedachten bezig met games, ook als je niet kunt spelen | peripheral | 23.6 |
| vat7 | preoccupation (cognitive salience) | you look forward to the next time you can game | Je verheugt je steeds op de volgende keer dat je kunt gamen | peripheral | 41.5 |
| vat8 | conflict (intra-personal) | you regularly think you should be gaming less often | Je denkt regelmatig dat je eigenlijk minder zou moeten gamen | core | 9.7 |
| vat9 | loss of control (failed attempts to stop) | you tried to spend less time on gaming, but were unsuccessful | Je hebt geprobeerd om minder tijd aan gamen te besteden en dat is niet gelukt | core | 5.9 |
| vat10 | withdrawal symptoms | you feel restless, frustrated, or irritated when you cannot game | Je voelt je telkens rot wanneer je niet kunt gamen | core | 4.0 |
| vat11 | problems (work/academic) | you rush through your (home)work to play games | Je raffelt je (huis)werk vaak af om te kunnen gamen | core | 9.9 |
| vat12 | problems (work/academic) | you neglect to do your (home)work because you prefer to game | Je doet je (huis)werk regelmatig niet omdat je wilt gamen | core | 9.0 |
| vat13 | mood modification/escapism | you often game because you are feeling down | Je gaat regelmatig gamen omdat je je rot voelt | peripheral | 25.9 |
| vat14 | mood modification/escapism | you often game to forget about a problem | Je gaat vaak gamen om een probleem te vergeten | peripheral | 30.2 |

**Table 6.** Items wordings for the Clinical Video Game Addiction Test 2.0 [51,89]. The canonical CVAT variants are in Dutch. English translations provided are not the result of a formal translation process; these are for reference only and may diverge slightly from the English translations in the original paper.

| label in data | construct | English wording | Dutch wording | core or peripheral | per cent endorsed |
|---|---|---|---|---|---|
| cvidat1 | tolerance | you continue to spend more and more time playing games | Je bent steeds meer tijd gaan besteden aan het spelen van games | peripheral | 32.5 |
| cvidat2 | withdrawal symptoms | you feel stressed, annoyed, or angry if you could not play games | Je voelt je gestrest, geïrriteerd, of boos als je niet kunt gamen | core | 8.0 |
| cvidat3 | loss of control (failed attempts to stop) | you are unable to play less, even though you want to | Het lukt je niet om minder te gaan gamen, terwijl je dat wel wilt | core | 6.4 |
| cvidat4 | preoccupation (cognitive salience) | you can hardly think about anything else than playing games when you are not gaming | Je kunt nauwelijks aan iets anders denken dan aan gamen als je niet aan het gamen bent | peripheral | 3.8 |
| cvidat5 | mood modification/ escapism | you regularly play videogames to avoid thinking about problems (difficulties) | Je gaat regelmatig gamen om niet aan problemen (moeilijkheden) te hoeven denken | peripheral | 32.1 |
| cvidat6 | behavioural salience (loss of interests) | you regularly neglect important activities because of gaming (such as social connections, school/work, or other hobbies) | Je verwaarloost regelmatig belangrijke zaken of personen om te kunnen gamen (zoals sociale contacten, school, werk, andere hobby's) | core | 11.3 |
| cvidat7 | lying/deception | you sometimes lie to others about the amount of time you spend gaming | Je liegt wel eens tegen anderen over hoe lang je aan het gamen bent geweest | core | 15.1 |
| cvidat8 | continuation despite problems | you still played games even though you knew this was causing problems with your family, friends, at work, or at school | Je hebt toch games gespeeld terwijl je wist dat dat problemen gaf met je familie, vrienden, werk of op school | core | 23.3 |
| cvidat9 | problems (work/ social) | you have had to give up or greatly reduce important activities because of gaming, such as sports, work, or interacting with friends and family | Je hebt door het gamen belangrijke activiteiten moeten opgeven of sterk verminderen - zoals sport, werk, of omgaan met vrienden en familie | core | 7.8 |
| cvidat10[a] | craving | you have a strong urge (need) to play video games | Je ervaart vaak sterke drang (behoefte) om te gaan gamen | peripheral | 14.2 |

(*Continued.*)

**Table 6.** (*Continued.*)

| label in data | construct | English wording | Dutch wording | core or peripheral | per cent endorsed |
|---|---|---|---|---|---|
| cvidat11[a] | problems (health) | you neglect your own health because of gaming (examples: not getting enough sleep, showering less or failing to brush teeth, eating worse, drinking insufficiently) | Je verwaarloost je eigen gezondheid door het gamen (zoals slaaptekort, minder douchen of tanden poetsen, slechter eten, niet voldoende drinken) | core | 12.7 |

[a]cvidat10 and cvidat11 are only included in the 11-item version of the CVAT-2

**Table 7.** Items wordings for the ICD-11 (novel scale based on definition of gaming disorder in the ICD-11; [3]). The ICD-11 measure was drafted in Dutch based on the English wording of the ICD-11. The English translations are not the result of a formal translation process and are provided simply as a reference for the reader.

| label in data | construct | English wording | Dutch wording | core or peripheral | per cent endorsed |
|---|---|---|---|---|---|
| icd111 | loss of control | you are unable to control your gaming (e.g. when you start gaming, how often and in which situations). | Het lukt je niet om gamen onder controle te houden (bijv. wanneer je met gamen begint, hoe vaak je gamet en in welke situaties je gaat gamen) | core | 9.7 |
| icd112 | behavioural salience (loss of interests) | you continuously give gaming priority over other hobbies and daily activities | Je geeft gamen de hele tijd voorrang boven andere hobby's en dagelijkse activiteiten | core | 16.0 |
| icd113 | continuation despite problems | you continue gaming even when this is creating problems | Je blijft gamen terwijl dit problemen oplevert | core | 8.3 |
| icd114 | sustained problems | gaming has been causing you problems for over a year. | Gamen levert al langer dan een jaar problemen op voor je | core | 5.4 |

**Table 8.** Items wordings for the ODBA scale (novel scale based on the open definition of behavioural addiction; [5,54]). The ODBA measure was drafted in Dutch based on the English wording of the ODBA. The English translations are not the result of a formal translation process and are provided simply as a reference for the reader.

| label in data | construct | English wording | Dutch wording | core or peripheral | per cent endorsed |
|---|---|---|---|---|---|
| odba1 | problems (general) | gaming creates various problems and stress in your life | Gamen veroorzaakt allerlei problemen en stress in je leven | core | 2.8 |
| odba2 | sustained problems | your gaming has been causing problems for you for months, but you are unable to play less. | Je hebt al maanden last van problemen door het gamen, maar het lukt je niet om te minderen | core | 1.9 |
| odba3[a] | automaticity | the time you spend on games is a conscious choice | Je tijdsbesteding aan games is een bewuste keuze | peripheral | 4.0 |
| odba4[a] | problems (general) | gaming does not lead to problems in your daily life (such as with school, work, other interests, friends, or family) | Gamen leidt niet tot problemen in je dagelijks leven (zoals school, werk, andere interesses, familie of vrienden) | core | 12.5 |

[a]Reverse scored.

**Table 9.** Items wordings for the Global Kids Online scale (http://globalkidsonline.net/tools/survey/). In contrast to the above measures, the English language version of the GKO is the canonical one as found in the original survey materials. Dutch wordings were drafted using a triple translate–retranslate–check process.

| label in data | construct | English wording | Dutch wording | core or peripheral | per cent endorsed |
|---|---|---|---|---|---|
| gko1 | problems (health) | you have gone without eating or sleeping because of the time you spent on gaming | Je mist maaltijden of slaap door de tijd die je met gamen doorbrengt | core | 9.7 |
| gko2 | conflict (interpersonal) | you have experienced conflicts with family or friends because of the time you spent on gaming | Je hebt conflicten met familie en vrienden door de tijd die je aan het gamen besteedt | core | 3.8 |
| gko3 | problems (academic) | Your grades [or work performance] have dropped because of the time you spent on gaming | Op werk of school presteer je slechter door de tijd die je besteedt aan gamen | core | 6.4 |

(Continued.)

**Table 9.** (*Continued.*)

| label in data | construct | English wording | Dutch wording | core or peripheral | per cent endorsed |
|---|---|---|---|---|---|
| gko4 | loss of control (failed attempts to stop) | you have tried unsuccessfully to spend less time on gaming | Je probeert om minder tijd te gamen, maar dit lukt niet | core | 4.5 |
| gko5 | problems (general) | you think the amount of time you spend on gaming causes problems for you | Je komt in de problemen door de tijd die je aan gamen besteedt | core | 4.5 |

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
