## [Peer Review File · Royal Society Open Science]

Review History

RSOS-201385.R0 (Original submission)

Review form: Reviewer 1

Is the manuscript scientifically sound in its present form?

Yes

Are the interpretations and conclusions justified by the results?

Yes

Is the language acceptable?

Yes

Do you have any ethical concerns with this paper?

No

Have you any concerns about statistical analyses in this paper?

Yes

Recommendation?

Accept with minor revision (please list in comments)

Comments to the Author(s)

I think the present manuscript is an excellent addition to the immense literature on problematic gaming, and a very suitable application for a specification curve analysis. I think the manuscript is highly accessible, well written, the analyses sound and the conclusions well-grounded in the empirical findings. I also checked the data and code and found them to be in very good shape. I only have one semi-major and some minor concerns/comments:

Major concern: You report yourselves that filling out the repeated pages of very similar items is a brutal task, and I am concerned that this will introduce a lot of communalities between the measures, either from straightlining or satisficing, or other stuff. Were the measures rotated, or did you include other precautions in the survey? And instead of discarding 200+ incomplete responses, I would encourage you to take another look not only at sociodemographic differences, but also the relevant dysregulated gaming measures, if possible. At least some should be available to check the correlations between them, etc.

Minor issues:

1. You report SE and CI for the Median estimate, but it is not completely clear where the SE and CI come from, i.e. what sampling distribution is underlying these estimates. Not saying it's wrong, but please check that the reported SE/CI make sense.
2. Your report "The median completion time for the survey was 812 minutes." , and I really hope that is a typo :-)
3. I think Figure 1 would be more helpful to the reader with actual correlation scores instead of ** all over the place, even if the figure is larger or the font size smaller. You could also consider removing the covariates to get more space.
4. Maybe I did not see it explicitly mentioned: You used mean scores for the gaming measures, not sum scores, right? Because on p. 23 you state "i.e. the sum score of gaming disorder measures."
5. Figure 4 needs a `+theme_bw()` to match the other figures.

Review form: Reviewer 2

Is the manuscript scientifically sound in its present form?

Yes

Are the interpretations and conclusions justified by the results?

Yes

Is the language acceptable?

Yes

Do you have any ethical concerns with this paper?

No

Have you any concerns about statistical analyses in this paper?

No

Recommendation?

Accept with minor revision (please list in comments)

Comments to the Author(s)

I was given the opportunity to review the manuscript, *The relationship between dysregulated gaming and mental well-being: A specification curve analysis of five gaming disorder scales*. The present study examined the differences in the relationship between problem video gaming and mental well-being based on the selection of mental well-being and problem video gaming measures, covariates, and samples. The paper presents a novel statistical approach for thoroughly examining the analytical flexibility of this relationship.

Literature review

The literature review was very concise and thorough. No comments.

Methods/Results

At the bottom of page 6, I believe there is a missing decimal as I do not believe participants took an average of more than 13 hours to complete the survey, correct?

I agree with the use of McDonald's hierarchical omega in lieu of Cronbach's alpha, but could the authors provide a measure of reliability for their other measures (i.e., gaming motivations, need satisfaction, mental well-being measures).

Given the novelty of their analyses, the authors might want to include a mention about statistical power and family-wise error in the context of specification curve analyses.

Discussion

In their discussion of the Pathways Model (Blaszczynski & Nower, 2002 [already cited]), I would suggest including the a recent application to video gaming by Lee, Lee, and Choo (2016).

There is ample evidence and theory to support daily need frustration as a risk factor for problem video gaming above and beyond both daily need satisfaction and need satisfaction during video gaming (see Vansteenkiste & Ryan, 2013; Mills & Allen, 2020; Mills et al., 2018 [already cited]). I believe this needs to be addressed as a limitation of the study, even if it was not the primary focus of the study, as it highlights the role of the social environment in pushing individuals to internalizing a more problematic pattern of play.

Within two sections the authors discuss the inadvertent assessment of the presence of negative affect while assessing the mood modification criteria (line 613-623) and escapism motivation (line 663-673). An excellent observation for the field to consider but have we not seen evidence of time invariance with regard to these measures (Chen et al., 2020; Stavropoulos, Bamford, Beard, Gomez, & Griffiths, 2019)? The notion that mood modification assesses the presence of negative mood might suggest that we would then see some changes over time. At this point, does the evidence suggest a lack of change over time? As for motivation, I am not aware of a multi-wave study examining the fluidity of motivation. But it would be expected that negative affect might influence individuals use of video gaming as an escape, no?

The use of the Digital Games Motivation Scale is not the most widely used measure. Could the authors comment on the differences between this measure and others such as the Motives for Online Gaming Questionnaire (Demetrovics et al., 2011) or Yee's (2006) components of gaming motives?

Referenced Citations in Review

Chen, I.-H., Strong, C., Lin, Y.-C., Tsai, M.-C., Leung, H., Lin, C.-Y., ... Griffiths, M. D. (2020). Time invariance of three ultra-brief internet-related instruments: Smartphone Application-Based Addiction Scale (SABAS), Bergen Social Media Addiction Scale (BSMAS), and the nine-item Internet Gaming Disorder Scale- Short Form (IGDS-SF9) (Study Part B). *Addictive Behaviors*, 101(November 2018), 105960. <https://doi.org/10.1016/j.addbeh.2019.04.018>

Demetrovics, Z., Urbán, R., Nagygyörgy, K., Farkas, J., Zilahy, D., Mervó, B., ... Harmath, E. (2011). Why do you play? The development of the motives for online gaming questionnaire (MOGQ). *Behavior Research Methods*, 43(3), 814–825. <https://doi.org/10.3758/s13428-011-0091-y>

Lee, S.-Y., Lee, H. K., & Choo, H. (2016). Typology of Internet gaming disorder and its clinical implications. *Psychiatry and Clinical Neurosciences*, 1–13. <https://doi.org/10.1111/pcn.12457>

Mills, D. J., & Allen, J. J. (2020). Self-determination theory, internet gaming disorder, and the mediating role of self-control. *Computers in Human Behavior*, 105(August 2019), 106209. <https://doi.org/10.1016/j.chb.2019.106209>

Stavropoulos, V., Bamford, L., Beard, C., Gomez, R., & Griffiths, M. D. (2019). Test-Retest Measurement Invariance of the Nine-Item Internet Gaming Disorder Scale in Two Countries: A Preliminary Longitudinal Study. *International Journal of Mental Health and Addiction*. <https://doi.org/10.1007/s11469-019-00099-w>

Vansteenkiste, M., & Ryan, R. M. (2013). On psychological growth and vulnerability: Basic psychological need satisfaction and need frustration as a unifying principle. *Journal of Psychotherapy Integration*, 23(3), 263–280. <https://doi.org/10.1037/a0032359>

Yee, N. (2006). Motivations for play in online games. *Cyberpsychology and Behavior*, 9(6), 772–775. <https://doi.org/10.1089/cpb.2006.9.772>

Review form: Reviewer 3

Is the manuscript scientifically sound in its present form?

Yes

Are the interpretations and conclusions justified by the results?

Yes

Is the language acceptable?

Yes

Do you have any ethical concerns with this paper?

Yes

Have you any concerns about statistical analyses in this paper?

No

Recommendation?

Accept with minor revision (please list in comments)

Comments to the Author(s)

This is an exceptionally novel and strong manuscript that addresses a host of difficult questions in the field using rigorous analytic techniques. I am very glad to see an improvement on Orben and Przybylski that is specific to gaming. The level of clarity and explanation in this manuscript is refreshing and I applaud the authors on their attention to transparency. While I think the manuscript is publishable, it is a complex analysis and writeup that I think would be improved with clarification of a few details. As I ask for every paper I review, I would also ask that you consider using the STROBE checklist (included as attachment) to promote consistent reporting of observational studies. A quick check shows that most items are already done, but checking this over and including it in the OSF project website will further support the open science aspects. You might also want to include the OSF link in the body of the paper. I saw only the DataVerse link and missed the OSF until I went back to the beginning and end bits.

A few concerns might warrant further attention. First, little data is provided about the sample other than demographics and the fact that it was recruited from different websites. The DataVerse page makes it seem as if the sample came from only the two websites listed. If this is the case, further explanation of the differences between the two samples should be included, as this might drive some of the unexpected findings. Also, the treatment of missing data should be expanded on.

Second, I see from a footnote in the original SCA paper that cases with missing data confuse the analysis process. Given that linear regression is chosen as the model, how does inclusion of only subjects with complete data challenge assumptions? A brief sentence about whether this adds limitations is warranted. Since SCA is a new and complex type of analysis, a few additional sentences explaining it – and especially explaining how to interpret the results – would be very useful. The comparisons with Orben & Przybylski are very useful, so thank you for that. I still find it difficult to interpret the figure especially.

Third, there are some surprising inconsistencies in responses to similar items as illustrated in Table 4, e.g. Continuation despite problems. It seems especially strange that the endorsement of icd113 continuation is only 8.3% when this seems to be a much broader criterion than cvidat8 (endorsed by 23.3%). I notice that icd113 is missing something like "voor je", which is present in icd114. I also note that cvid8 gives examples. I'm wondering what the authors think is the reason for this differential endorsement. In any case, it would be good to better understand what might account for the differences.

Also, I'm not sure if the chosen colors are easily understood by people with colorblindness-this might be something to consider as well. Another small thing is that Orben and Przybylski is referenced a lot with "technology use"; it might be good to quickly mention in the intro what that study was and how they operationalized tech use

Specific recommendations follow.

Abstract

Include sampling method

"identify a maximally parsimonious..." this phrase is somewhat hard to understand.

Methods (by number)

- 168: People outside of Europe might not know what Flemish subjects are, so you could consider adding "Dutch-speaking people of Belgium" or some such.
Also, please consider describing more about the sampling procedure, websites, who might visit them, etc., the total number of survey questions, and whether survey questions were randomized.
- 179: Average time on the DataVerse page differs from median time by almost 800 minutes; is this correct?
- 190: Were any other aspects compared, e.g. where the respondent was recruited from?
- 201: Consider adding that reliability measures for each scale appear in Table 4.
- Dysregulated gaming: Consider adding averages/average endorsement percentages for clinical populations where available just to give context for the sample's average/endorsement percentages.
- 246: Consider adding year to "the time that this study was conceptualized". Might be worth discussing how the criteria used compare to what was eventually adopted, either here or in Discussion.
- 270: Please add a link or references to the ODBA on OSF.
- 283-4: Please explain why the final item is divergent.
- 331: I could not find the exact numbers in the supplementary materials because I saw only the DataVerse referenced in the table note; perhaps insert the OSF link here.
- 334: Please add 1-2 more sentences that describe SCA, its assumptions and how it treats missing data. Also, are there corrects for multiple comparisons, or is this not necessary in this approach?
- 372: Why are combinations of covariates not part of the specifications? Is this not possible?

Results (by line number)

- 398: How does McDonald's omega compare to alpha? Is a "good" value also $>.8$?
- 423: VAT was fine as a single factor when it was developed, right? Some explanation of why it doesn't seem to hold together in this sample might be good.
- 445: What are the possible reasons for the low association between ODBA and wellbeing? Could this be related to the divergent item wording?

Figure 2: This could use a bit more explanation. What is being regressed on what? Do the lines correspond to points in Orben & Przybylski? I'm still not sure what the models are ranked by. I would also suggest using "control variables" rather than "controls".

Table 3: Please include outcome and predictor variables or concepts (e.g., regressions of dysregulated gaming on wellbeing) in the table title and/or notes.

- 464: Do you have a citation for the decomposed variance approach? Or is it part of the original SCA paper?
- 474: What a huge change! I realize this is in the discussion, but something like "suggesting...." would be good here so it can be immediately interpreted.

Figure 3: "Value" is not easily interpreted as an axis title. I'm not sure how Fig 3 relates to Fig 2. It seems like perhaps the distributions are not for each analysis choice but for all analyses in which they appear?

Fig 4: Perhaps add briefly why ODBA is excluded

492: for the explanatory variable of...? Include concept here

Table 4: The difference in effect sizes between similar items is surprising, e.g. Continuation despite problems. I see the percent endorsing these items is also different. This might benefit from further explanation and inclusion of percent endorsing the item.

Discussion (by line number)

How could the fact that all models yielded significant results relate to the different subgroups of the sample or the recruitment method?

510: Typo-"flexibility's" □ flexibilities

557: Did Gentile even measure depression and anxiety as possible causes of PG in their longitudinal model? This is a problem with the conceptualization of causality in this body of literature and could be mentioned.

641: Why might the full versions not be unidimensional? Wouldn't this have appeared in prior research? It might be good to clarify what might be different here.

652: "in unique ways" is not very clear. Your earlier suggestion was that there might be a confounding factor causing gaming for escape, right?

728: Please discuss the weaknesses and assumptions of the approaches, and items with the same concept (e.g., continuation despite problems) might receive such different endorsements from 2 scales.

735: "was attributable to the choice of dysregulated gaming measure" - wasn't this once ODBA was dropped?

738: missing "to" in "noise the association"

Appendix

How were the items translated to English? I wonder whether the translations from English ICD-11 to Dutch might be responsible for the difference between the similar Table 4 items and how they were endorsed here.

Here are some native-English speaker/Dutch language learner suggestions that could be considered, either as things to explain some of the surprising findings or for future use:

VAT vs CVAT mood modification/withdrawal: "rot" is defined differently in Dutch and English between the questions that use it (or don't use it, i.e. cvidat2 Withdrawal, which specifies the feelings clearly)

vat14: English has plural "problems" while Dutch has "a problem"; civdat5 does have the plural

vat4: English translation is missing "often"; without that there is a sense of "always" or "never" prefer, perhaps

cvidat2: English wording adds "were not allowed"

cvidat8: "You played games" doesn't convey the same flavor of continuing as "je hebt toch games gespeeld"

cvidat9: gaming appears twice in the English translation

icd112: I don't think "other life interests" translates to "hobby's", but is meant to include all daily activities including functional roles like worker, student, family member.

icd113: Does the Dutch imply "for you"?

icd114: English "problematic" is different from Dutch "creating problems". I would suggest the difference between the wording of this question and the endorsement of a long syndrome of symptoms and problems be brought out in the Discussion. (e.g., it's not just that gaming is causing problems, but that the combination of symptoms and problems has persisted).

odba2: I'm not sure if the "om te minderen" could be misinterpreted as "problemen om te minderen" or if gaming is clearly implied

odba3: Why was this criteria divergent from what was in the OSF definition?

Decision letter (RSOS-201385.R0)

Dear Mr Ballou,

On behalf of the Editors, we are pleased to inform you that your Manuscript RSOS-201385 "The relationship between dysregulated gaming and mental well-being: A specification curve analysis of five gaming disorder scales" has been accepted for publication in Royal Society Open Science subject to minor revision in accordance with the referees' reports. Please find the referees' comments along with any feedback from the Editors below my signature.

Please submit your revised manuscript and required files (see below) no later than 7 days from today's (ie 21-Oct-2020) date. Note: the ScholarOne system will 'lock' if submission of the revision is attempted 7 or more days after the deadline. If you do not think you will be able to meet this deadline please contact the editorial office immediately.

on behalf of Dr Christina Demski (Associate Editor) and Essi Viding (Subject Editor)
openscience@royalsociety.org

Associate Editor Comments to Author (Dr Christina Demski):

Both reviewers see value in this manuscript but highlight a few methodological and conceptual issues which they would like to more fully address in a revision before it can be accepted and published.

Reviewer comments to Author:

Reviewer: 1

Comments to the Author(s)

I think the present manuscript is an excellent addition to the immense literature on problematic gaming, and a very suitable application for a specification curve analysis. I think the manuscript is highly accessible, well written, the analyses sound and the conclusions well-grounded in the empirical findings. I also checked the data and code and found them to be in very good shape. I only have one semi-major and some minor concerns/comments:

Major concern: You report yourselves that filling out the repeated pages of very similar items is a brutal task, and I am concerned that this will introduce a lot of communalities between the measures, either from straightlining or satisficing, or other stuff. Were the measures rotated, or did you include other precautions in the survey? And instead of discarding 200+ incomplete responses, I would encourage you to take another look not only at sociodemographic differences, but also the relevant dysregulated gaming measures, if possible. At least some should be available to check the correlations between them, etc.

Minor issues:

1. You report SE and CI for the Median estimate, but it is not completely clear where the SE and CI come from, i.e. what sampling distribution is underlying these estimates. Not saying it's wrong, but please check that the reported SE/CI make sense.
2. Your report "The median completion time for the survey was 812 minutes." , and I really hope that is a typo :-)
3. I think Figure 1 would be more helpful to the reader with actual correlation scores instead of ** all over the place, even if the figure is larger or the font size smaller. You could also consider removing the covariates to get more space.
4. Maybe I did not see it explicitly mentioned: You used mean scores for the gaming measures, not sum scores, right? Because on p. 23 you state "i.e. the sum score of gaming disorder measures."
5. Figure 4 needs a `+theme_bw()` to match the other figures.

Reviewer: 2

Comments to the Author(s)

I was given the opportunity to review the manuscript, The relationship between dysregulated gaming and mental well-being: A specification curve analysis of five gaming disorder scales. The present study examined the differences in the relationship between problem video gaming and mental well-being based on the selection of mental well-being and problem video gaming measures, covariates, and samples. The paper presents a novel statistical approach for thoroughly examining the analytical flexibility of this relationship.

Literature review

The literature review was very concise and thorough. No comments.

Methods/Results

At the bottom of page 6, I believe there is a missing decimal as I do not believe participants took an average of more than 13 hours to complete the survey, correct?

I agree with the use of McDonald's hierarchical omega in lieu of Cronbach's alpha, but could the authors provide a measure of reliability for their other measures (i.e., gaming motivations, need satisfaction, mental well-being measures).

Given the novelty of their analyses, the authors might want to include a mention about statistical power and family-wise error in the context of specification curve analyses.

Discussion

In their discussion of the Pathways Model (Blaszczynski & Nower, 2002 [already cited]), I would suggest including the a recent application to video gaming by Lee, Lee, and Choo (2016).

There is ample evidence and theory to support daily need frustration as a risk factor for problem video gaming above and beyond both daily need satisfaction and need satisfaction during video gaming (see Vansteenkiste & Ryan, 2013; Mills & Allen, 2020; Mills et al., 2018 [already cited]) . I believe this needs to be addressed as a limitation of the study, even if it was not the primary focus of the study, as it highlights the role of the social environment in pushing individuals to internalizing a more problematic pattern of play.

Within two sections the authors discuss the inadvertent assessment of the presence of negative affect while assessing the mood modification criteria (line 613-623) and escapism motivation (line 663-673). An excellent observation for the field to consider but have we not seen evidence of time invariance with regard to these measures (Chen et al., 2020; Stavropoulos, Bamford, Beard, Gomez, & Griffiths, 2019)? The notion that mood modification assesses the presence of negative mood might suggest that we would then see some changes over time. At this point, does the evidence suggest a lack of change over time? As for motivation, I am not aware of a multi-wave study examining the fluidity of motivation. But it would be expected that negative affect might influence individuals use of video gaming as an escape, no?

The use of the Digital Games Motivation Scale is not the most widely used measure. Could the authors comment on the differences between this measure and others such as the Motives for Online Gaming Questionnaire (Demetrovics et al., 2011) or Yee's (2006) components of gaming motives?

Referenced Citations in Review

Chen, I.-H., Strong, C., Lin, Y.-C., Tsai, M.-C., Leung, H., Lin, C.-Y., ... Griffiths, M. D. (2020). Time invariance of three ultra-brief internet-related instruments: Smartphone Application-Based Addiction Scale (SABAS), Bergen Social Media Addiction Scale (BSMAS), and the nine-item Internet Gaming Disorder Scale- Short Form (IGDS-SF9) (Study Part B). *Addictive Behaviors*, 101(November 2018), 105960. <https://doi.org/10.1016/j.addbeh.2019.04.018>

Demetrovics, Z., Urbán, R., Nagygyörgy, K., Farkas, J., Zilahy, D., Mervó, B., ... Harmath, E. (2011). Why do you play? The development of the motives for online gaming questionnaire (MOGQ). *Behavior Research Methods*, 43(3), 814-825. <https://doi.org/10.3758/s13428-011-0091-y>

Lee, S.-Y., Lee, H. K., & Choo, H. (2016). Typology of Internet gaming disorder and its clinical implications. *Psychiatry and Clinical Neurosciences*, 1–13. <https://doi.org/10.1111/pcn.12457>

Mills, D. J., & Allen, J. J. (2020). Self-determination theory, internet gaming disorder, and the mediating role of self-control. *Computers in Human Behavior*, 105(August 2019), 106209. <https://doi.org/10.1016/j.chb.2019.106209>

Stavropoulos, V., Bamford, L., Beard, C., Gomez, R., & Griffiths, M. D. (2019). Test-Retest Measurement Invariance of the Nine-Item Internet Gaming Disorder Scale in Two Countries: A Preliminary Longitudinal Study. *International Journal of Mental Health and Addiction*. <https://doi.org/10.1007/s11469-019-00099-w>

Vansteenkiste, M., & Ryan, R. M. (2013). On psychological growth and vulnerability: Basic psychological need satisfaction and need frustration as a unifying principle. *Journal of Psychotherapy Integration*, 23(3), 263–280. <https://doi.org/10.1037/a0032359>

Yee, N. (2006). Motivations for play in online games. *Cyberpsychology and Behavior*, 9(6), 772–775. <https://doi.org/10.1089/cpb.2006.9.772>

Reviewer: 3

Comments to the Author(s)

Note: This review is also included as an attachment if that's easier to work with.

This is an exceptionally novel and strong manuscript that addresses a host of difficult questions in the field using rigorous analytic techniques. I am very glad to see an improvement on Orben and Przybylski that is specific to gaming. The level of clarity and explanation in this manuscript is refreshing and I applaud the authors on their attention to transparency. While I think the manuscript is publishable, it is a complex analysis and writeup that I think would be improved with clarification of a few details. As I ask for every paper I review, I would also ask that you consider using the STROBE checklist (included as attachment) to promote consistent reporting of observational studies. A quick check shows that most items are already done, but checking this over and including it in the OSF project website will further support the open science aspects. You might also want to include the OSF link in the body of the paper. I saw only the DataVerse link and missed the OSF until I went back to the beginning and end bits.

A few concerns might warrant further attention. First, little data is provided about the sample other than demographics and the fact that it was recruited from different websites. The DataVerse page makes it seem as if the sample came from only the two websites listed. If this is the case, further explanation of the differences between the two samples should be included, as this might drive some of the unexpected findings. Also, the treatment of missing data should be expanded on.

Second, I see from a footnote in the original SCA paper that cases with missing data confuse the analysis process. Given that linear regression is chosen as the model, how does inclusion of only subjects with complete data challenge assumptions? A brief sentence about whether this adds limitations is warranted. Since SCA is a new and complex type of analysis, a few additional sentences explaining it – and especially explaining how to interpret the results – would be very useful. The comparisons with Orben & Przybylski are very useful, so thank you for that. I still find it difficult to interpret the figure especially.

Third, there are some surprising inconsistencies in responses to similar items as illustrated in Table 4, e.g. Continuation despite problems. It seems especially strange that the endorsement of

icd113 continuation is only 8.3% when this seems to be a much broader criterion than cvidat8 (endorsed by 23.3%). I notice that icd113 is missing something like "voor je", which is present in icd114. I also note that cvid8 gives examples. I'm wondering what the authors think is the reason for this differential endorsement. In any case, it would be good to better understand what might account for the differences.

Also, I'm not sure if the chosen colors are easily understood by people with colorblindness-this might be something to consider as well. Another small thing is that Orben and Przybylski is referenced a lot with "technology use"; it might be good to quickly mention in the intro what that study was and how they operationalized tech use

Specific recommendations follow.

Abstract

Include sampling method

"identify a maximally parsimonious..." this phrase is somewhat hard to understand.

Methods (by number)

168: People outside of Europe might not know what Flemish subjects are, so you could consider adding "Dutch-speaking people of Belgium" or some such.

Also, please consider describing more about the sampling procedure, websites, who might visit them, etc., the total number of survey questions, and whether survey questions were randomized.

179: Average time on the DataVerse page differs from median time by almost 800 minutes; is this correct?

190: Were any other aspects compared, e.g. where the respondent was recruited from?

201: Consider adding that reliability measures for each scale appear in Table 4.

Dysregulated gaming: Consider adding averages/average endorsement percentages for clinical populations where available just to give context for the sample's average/endorsement percentages.

246: Consider adding year to "the time that this study was conceptualized". Might be worth discussing how the criteria used compare to what was eventually adopted, either here or in Discussion.

270: Please add a link or references to the ODBA on OSF.

283-4: Please explain why the final item is divergent.

331: I could not find the exact numbers in the supplementary materials because I saw only the DataVerse referenced in the table note; perhaps insert the OSF link here.

334: Please add 1-2 more sentences that describe SCA, its assumptions and how it treats missing data. Also, are there corrects for multiple comparisons, or is this not necessary in this approach?

372: Why are combinations of covariates not part of the specifications? Is this not possible?

Results (by line number)

398: How does McDonald's omega compare to alpha? Is a "good" value also $>.8$?

423: VAT was fine as a single factor when it was developed, right? Some explanation of why it doesn't seem to hold together in this sample might be good.

445: What are the possible reasons for the low association between ODBA and wellbeing? Could this be related to the divergent item wording?

Figure 2: This could use a bit more explanation. What is being regressed on what? Do the lines correspond to points in Orben & Przybylski? I'm still not sure what the models are ranked by. I would also suggest using "control variables" rather than "controls".

Table 3: Please include outcome and predictor variables or concepts (e.g., regressions of dysregulated gaming on wellbeing) in the table title and/or notes.

464: Do you have a citation for the decomposed variance approach? Or is it part of the original SCA paper?

474: What a huge change! I realize this is in the discussion, but something like "suggesting...." would be good here so it can be immediately interpreted.

Figure 3: "Value" is not easily interpreted as an axis title. I'm not sure how Fig 3 relates to Fig 2. It seems like perhaps the distributions are not for each analysis choice but for all analyses in which they appear?

Fig 4: Perhaps add briefly why ODBA is excluded

492: for the explanatory variable of...? Include concept here

Table 4: The difference in effect sizes between similar items is surprising, e.g. Continuation despite problems. I see the percent endorsing these items is also different. This might benefit from further explanation and inclusion of percent endorsing the item.

Discussion (by line number)

How could the fact that all models yielded significant results relate to the different subgroups of the sample or the recruitment method?

510: Typo-"flexibility's" □ flexibilities

557: Did Gentile even measure depression and anxiety as possible causes of PG in their longitudinal model? This is a problem with the conceptualization of causality in this body of literature and could be mentioned.

641: Why might the full versions not be unidimensional? Wouldn't this have appeared in prior research? It might be good to clarify what might be different here.

652: "in unique ways" is not very clear. Your earlier suggestion was that there might be a confounding factor causing gaming for escape, right?

728: Please discuss the weaknesses and assumptions of the approaches, and items with the same concept (e.g., continuation despite problems) might receive such different endorsements from 2 scales.

735: "was attributable to the choice of dysregulated gaming measure" - wasn't this once ODBA was dropped?

738: missing "to" in "noise the association"

Appendix

How were the items translated to English? I wonder whether the translations from English ICD-11 to Dutch might be responsible for the difference between the similar Table 4 items and how they were endorsed here.

Here are some native-English speaker/Dutch language learner suggestions that could be considered, either as things to explain some of the surprising findings or for future use:

VAT vs CVAT mood modification/withdrawal: "rot" is defined differently in Dutch and English between the questions that use it (or don't use it, i.e. cvidat2 Withdrawal, which specifies the feelings clearly)

vat14: English has plural "problems" while Dutch has "a problem"; civdat5 does have the plural
vat4: English translation is missing "often"; without that there is a sense of "always" or "never" prefer, perhaps

cvidat2: English wording adds "were not allowed"

cvidat8: "You played games" doesn't convey the same flavor of continuing as "je hebt toch games gespeeld"

cvidat9: gaming appears twice in the English translation

icd112: I don't think "other life interests" translates to "hobby's", but is meant to include all daily activities including functional roles like worker, student, family member.

icd113: Does the Dutch imply "for you"?

icd114: English "problematic" is different from Dutch "creating problems". I would suggest the difference between the wording of this question and the endorsement of a long syndrome of symptoms and problems be brought out in the Discussion. (e.g., it's not just that gaming is causing problems, but that the combination of symptoms and problems has persisted).

odba2: I'm not sure if the "om te minderen" could be misinterpreted as "problemen om te minderen" or if gaming is clearly implied

odba3: Why was this criteria divergent from what was in the OSF definition?

===PREPARING YOUR MANUSCRIPT===

- one version identifying all the changes that have been made (for instance, in coloured highlight, in bold text, or tracked changes);
- a 'clean' version of the new manuscript that incorporates the changes made, but does not highlight them.

This version will be used for typesetting.

===PREPARING YOUR REVISION IN SCHOLARONE===

Please ensure that you include a summary of your paper at Step 2 'Type, Title, & Abstract'. This should be no more than 100 words to explain to a non-scientific audience the key findings of your

research. This will be included in a weekly highlights email circulated by the Royal Society press office to national UK, international, and scientific news outlets to promote your work.

Author's Response to Decision Letter for (RSOS-201385.R0)

See Appendix A.

RSOS-201385.R1 (Revision)

Review form: Reviewer 1

Is the manuscript scientifically sound in its present form?

Yes

Are the interpretations and conclusions justified by the results?

Yes

Is the language acceptable?

Yes

Do you have any ethical concerns with this paper?

No

Have you any concerns about statistical analyses in this paper?

No

Recommendation?

Accept as is

Comments to the Author(s)

I think the authors adequately addressed the remaining issues raised by the reviewers and recommend publication.

Review form: Reviewer 3

Is the manuscript scientifically sound in its present form?

Yes

Are the interpretations and conclusions justified by the results?

Yes

Is the language acceptable?

Yes

Do you have any ethical concerns with this paper?

No

Have you any concerns about statistical analyses in this paper?

No

Recommendation?

Accept with minor revision (please list in comments)

Comments to the Author(s)

Thank you for attending to my multiple suggestions for revisions. I am satisfied with all of them. I do recommend some very minor things for clarity and to emphasize the impact of the findings.

If possible, could the many new findings and recommendations be listed in a highlights section or figure? There are so many implications for further research that I think this would be very useful.

p. 6

line 183 - the 87% is of what?

line 193 - sample size of 424 perhaps?

p. 7 line 261 - has the ADM been used in adults? How might the questions not apply?

p. 8 line 277 - contains one item - add "for each criterion"?

p. 14 Table 3 - I suggest you add something like "bivariate linear" to the "summary of regression coefficients" to the title

Figure 2 references dots, but I am unable to make out any dots in the figure. I think perhaps the top graph is simply too clustered, but I'm not sure.

p. 19 line 680 - important implications like this and the many others could be highlighted in a box at the beginning of the paper

p. 20

line 707 - this is really important as well and could be highlighted

line 719 - could mention the need to explore confounding (and highlight)

line 735 - another important implication

p. 85 Table A4-should the caption read the ODBA measure?

I would also like to suggest that should the authors pursue future studies with online recruiting, they should consider using non gaming-specific incentives such as bol.com or coolblue gift cards. Using a gaming-specific incentive could make more casual gamers less likely to participate.

Decision letter (RSOS-201385.R1)

Dear Mr Ballou

On behalf of the Editors, we are pleased to inform you that your Manuscript RSOS-201385.R1 "The relationship between dysregulated gaming and mental well-being: A specification curve analysis of five gaming disorder scales" has been accepted for publication in Royal Society Open Science subject to minor revision in accordance with the referees' reports. Please find the referees' comments along with any feedback from the Editors below my signature.

Please submit your revised manuscript and required files (see below) no later than 7 days from today's (ie 12-Jan-2021) date. Note: the ScholarOne system will 'lock' if submission of the revision is attempted 7 or more days after the deadline. If you do not think you will be able to meet this deadline please contact the editorial office immediately.

on behalf of Dr Christina Demski (Associate Editor) and Essi Viding (Subject Editor)
openscience@royalsociety.org

Reviewer comments to Author:
Reviewer: 1

Comments to the Author(s)
I think the authors adequately addressed the remaining issues raised by the reviewers and recommend publication.

Reviewer: 3
Comments to the Author(s)
Thank you for attending to my multiple suggestions for revisions. I am satisfied with all of them. I do recommend some very minor things for clarity and to emphasize the impact of the findings.

If possible, could the many new findings and recommendations be listed in a highlights section or figure? There are so many implications for further research that I think this would be very useful.

p. 6
line 183 - the 87% is of what?
line 193 - sample size of 424 perhaps?

p. 7 line 261 - has the ADM been used in adults? How might the questions not apply?

p. 8 line 277 - contains one item - add "for each criterion"?

p. 14 Table 3 - I suggest you add something like "bivariate linear" to the "summary of regression coefficients" to the title

Figure 2 references dots, but I am unable to make out any dots in the figure. I think perhaps the top graph is simply too clustered, but I'm not sure.

p. 19 line 680 - important implications like this and the many others could be highlighted in a box at the beginning of the paper

p. 20

line 707 - this is really important as well and could be highlighted

line 719 - could mention the need to explore confounding (and highlight)

line 735 - another important implication

p. 85 Table A4-should the caption read the ODBA measure?

I would also like to suggest that should the authors pursue future studies with online recruiting, they should consider using non gaming-specific incentives such as bol.com or coolblue gift cards. Using a gaming-specific incentive could make more casual gamers less likely to participate.

===PREPARING YOUR MANUSCRIPT===

===PREPARING YOUR REVISION IN SCHOLARONE===

Author's Response to Decision Letter for (RSOS-201385.R1)

See Appendix B.

Decision letter (RSOS-201385.R2)

Dear Mr Ballou,

It is a pleasure to accept your manuscript entitled "The relationship between dysregulated gaming and mental well-being: A specification curve analysis of five gaming disorder scales" in its current form for publication in Royal Society Open Science.

on behalf of Dr Christina Demski (Associate Editor) and Essi Viding (Subject Editor)
openscience@royalsociety.org

Appendix A

We would first like to sincerely thank all three reviewers for thoughtful and detailed feedback during a busy and stressful time for many. We address each comment in-line below.

Reviewer 1

Major concern: You report yourselves that filling out the repeated pages of very similar items is a brutal task, and I am concerned that this will introduce a lot of communalities between the measures, either from straightlining or satisficing, or other stuff. Were the measures rotated, or did you include other precautions in the survey?

The order of the measures was not randomized. This decision was made because of concerns that participants would end up filling in long stretches of negatively-oriented and/or repetitive scales and because of the widely varying lengths of the survey measures; this was compounded by limitations in question block randomization in Limesurvey.

Instead, dysregulated gaming measures were each alternated with non-dysregulated gaming measures and free response questions to try and prevent monotony and protracted series of negative items/Likert responses, which we briefly describe on line 183. We also placed some of the longest measures (gaming motivations, basic needs) at the beginning of the survey to mitigate survey fatigue. Lastly, we included 2 attention checks and a number of reverse-coded items throughout the questionnaire; the vast majority of participants answered both attention checks correctly, and the reverse-coded items perform as expected in all scales in which they appear.

Manual inspection of the data showed minimal signs of straight-lining; where this did occur, these were typically the same participants who failed one or both attention checks, justifying the decision to exclude these. We do not observe any systematic change in mean scores or decreasing within-scale variance with measures later in the survey.

We acknowledge that possible order effects remain a limitation of the study, and welcome future work that seeks to address this. We have expanded upon the survey design in the method section [line 163-170] and also emphasized this point in the limitation section [line 743-750].

And instead of discarding 200+ incomplete responses, I would encourage you to take another look not only at sociodemographic differences, but also the relevant dysregulated gaming measures, if possible. At least some should be available to check the correlations between them, etc.

Thank you for the suggestion—this comment prompted closer inspection of the pattern of participant drop-out and interestingly, this revealed that repetitive dysregulated gaming measures did not play a major role in non-completion. Of the 289 participants who opened the survey but did not complete it, 190 did not complete even the first measure (gaming motivations). 263 dropped out before the second

dysregulated gaming measure (which is also prior to the first well-being measure, precluding their use in any of the paper's primary analyses). We have added this information to the manuscript for context [line 173-180].

We also conducted additional tests on the first dysregulated gaming measure (CVAT-2), for which we have the greatest amount of complete data. In addition to the previously reported difference in education level, both dropped participants and those that failed the attention check report lower education levels and higher dysregulated gaming scores. Separately, these are not statistically significant (at $p < .05$), but when the entire included sample is compared to participants dropped for either reason, they become significant. We report the results of these tests in the method section [line 200-207].

To make the analyses easier to interpret for the reader, we have elected to continue to analyze only the complete cases, ensuring that the sample size for each of the models is the same (with the exception of models that discard outliers identified based on improbable reported gametime as a justifiable analytical decision). However, as a robustness check, we have also conducted the same specification curve analysis with the full dataset including incomplete responses (where missing data is handled using pairwise deletion). These do not change the conclusions of the paper and have been added to the supplementary materials.

1. You report SE and CI for the Median estimate, but it is not completely clear where the SE and CI come from, i.e. what sampling distribution is underlying these estimates. Not saying it's wrong, but please check that the reported SE/CI make sense.

The reported SE/CI denote the uncertainty for the regression coefficient of that individual model/specification using a standard t-distribution. We have added an additional sentence to the caption of table 3 to clarify this.

2. Your report "The median completion time for the survey was 812 minutes." , and I really hope that is a typo :-) [echoed by reviewers 2 and 3]

This was indeed an error - 812 was the length of time in seconds. We now report the correct time in minutes.

3. I think Figure 1 would be more helpful to the reader with actual correlation scores instead of ** all over the place, even if the figure is larger or the font size smaller. You could also consider removing the covariates to get more space.

We were previously having trouble with plot formatting, but have replaced the asterisks with numbers and increased the size of the figure so that the numbers are legible.

4. Maybe I did not see it explicitly mentioned: You used mean scores for the gaming measures, not sum scores, right? Because on p. 23 you state "i.e. the sum score of gaming disorder measures."

We have corrected this to reflect the use of mean scores.

5. Figure 4 needs a +theme_bw() to match the other figures.

We have added this for consistency between figures.

Reviewer 2

At the bottom of page 6, I believe there is a missing decimal as I do not believe participants took an average of more than 13 hours to complete the survey, correct?

(see also reply to reviewer 1) This was indeed an error - 812 was the length of time in seconds. We now report the correct time in minutes.

I agree with the use of McDonald's hierarchical omega in lieu of Cronbach's alpha, but could the authors provide a measure of reliability for their other measures (i.e., gaming motivations, need satisfaction, mental well-being measures).

We have added reliability scores for the remaining measures to the table of descriptive statistics (table 1).

Given the novelty of their analyses, the authors might want to include a mention about statistical power and family-wise error in the context of specification curve analyses.

In this paper, we use a specification curve analysis as essentially a single test of the relationship between dysregulated gaming and well-being; because inferences are not made based on the (non-)significance of individual models in the curve, we do not correct for multiple tests (in our particular case, the point is moot; the median p-value across our models is $< .0001$ and all models would have likely remained significant after correction). Nonetheless, depending on the power of the test, one would expect to see a certain number of significant results in the specification curve even if there was no true effect. We have added a paragraph to explain this rationale in the analytical approach section [line 384-393].

In their discussion of the Pathways Model (Blaszczynski & Nower, 2002 [already cited]), I would suggest including the a recent application to video gaming by Lee, Lee, and Choo (2016).

This is a relevant and interesting paper addressing heterogeneity in gaming disorder, and we have added a sentence about it into the discussion [line 645].

There is ample evidence and theory to support daily need frustration as a risk factor for problem video gaming above and beyond both daily need satisfaction and need satisfaction during video gaming (see Vansteenkiste & Ryan, 2013; Mills & Allen, 2020; Mills et al., 2018 [already cited]). I believe this needs to be addressed as a limitation of the study, even if it was not the primary focus of the study, as it highlights the role of the social environment in pushing individuals to internalizing a more problematic pattern of play.

This is a good point and an interesting area of SDT-informed work on dysregulated gaming. We have added some detail about this work to the section of the discussion on covariates/basic needs [line 697].

Within two sections the authors discuss the inadvertent assessment of the presence of negative affect while assessing the mood modification criteria (line 613-623) and escapism motivation (line 663-673). An excellent observation for the field to consider

Thank you for this comment.

... but have we not seen evidence of time invariance with regard to these measures (Chen et al., 2020; Stavropoulos, Bamford, Beard, Gomez, & Griffiths, 2019)? ... The notion that mood modification assesses the presence of negative mood might suggest that we would then see some changes over time. At this point, does the evidence suggest a lack of change over time? As for motivation, I am not aware of a multi-wave study examining the fluidity of motivation. But it would be expected that negative affect might influence individuals use of video gaming as an escape, no?

The referenced studies demonstrate time invariance of the factor loadings/unidimensional structure of IGD measures that include escapism; without a more complicated model that controls for both negative affect and gaming disorder severity, these are not sufficient to demonstrate that escapism is not confounded. More longitudinal research is needed to address the fluidity of these items over time—we can only state on the basis of our data that we suspect some interplay between the two. We have added a sentence to the discussion to emphasize the speculative nature of this argument and the need for dedicated research on the topic [line 677-679].

The use of the Digital Games Motivation Scale is not the most widely used measure. Could the authors comment on the differences between this measure and others such as the Motives for Online Gaming Questionnaire (Demetrovics et al., 2011) or Yee's (2006) components of gaming motives?

The DGMS was used in this study both because of its good evidence of validity in a multi-country study and the existence of a Dutch language version of the scale. We expect strong intercorrelations between certain subscales of the DGMS and related constructs from other questionnaires (e.g., between DGMS performance and Yee's advancement), but motivations are not the primary focus of this paper and we prefer for readability not to dive too deep in the manuscript into what is a complicated topic with a large body of literature.

Reviewer 3

As I ask for every paper I review, I would also ask that you consider using the STROBE checklist (included as attachment) to promote consistent reporting of observational studies. A quick check shows that most items are already done, but checking this over and including it in the OSF project website will further support the open science aspects.

We have completed a STROBE checklist and have uploaded it to the OSF project page.

You might also want to include the OSF link in the body of the paper. I saw only the DataVerse link and missed the OSF until I went back to the beginning and end bits.

A link to the OSF page has now been added alongside the Dataverse link [line 434].

Little data is provided about the sample other than demographics and the fact that it was recruited from different websites. The DataVerse page makes it seem as if the sample came from only the two websites listed. If this is the case, further explanation of the differences between the two samples should be included, as this might drive some of the unexpected findings.

We have added additional detail to the participants and procedure section to clarify how and where the sample was recruited [line 155-162]. Specifically, the Dutch sample was recruited using a Facebook advertisement campaign; a minority of participants (49 before cleaning) joined the study from the public health website (gameninfo.nl) directly. The Flemish sample came entirely from the gaming journalism website.

We also conducted additional tests for differences between the Dutch and Flemish participants; results indicate that the Dutch participants reported slightly higher levels of dysregulation on average; this is likely an effect of the fact that Dutch participants (mean: 22 years) were on average younger than Flemish ones (mean: 28 years). We report the results of these tests on line 193.

Despite these differences, Given overall high similarity between the two groups (same language, high degree of cultural overlap, participants all self-selected due to interest in gaming) and no theoretical

reason to expect the relationship between dysregulation and well-being to vary between the two countries, we elected to analyze them together.

Also, the treatment of missing data should be expanded on [...] Second, I see from a footnote in the original SCA paper that cases with missing data confuse the analysis process. Given that linear regression is chosen as the model, how does inclusion of only subjects with complete data challenge assumptions? A brief sentence about whether this adds limitations is warranted.

Because we excluded incomplete responses and each item required a response, there is no missing data in the current analyses [mentioned on line 204-5 of the original manuscript, 192 of revised manuscript].

Under other circumstances, specification curve analyses can be used in combination with any standard method for dealing with missing data (e.g., pairwise deletion)—each model in the SCA would simply have a different sample size as a result, which can in case of large fluctuations be plotted alongside the curve (see bottom of <https://masurp.github.io/specr/articles/custom-plot.html> for an example). This is the approach taken in the added supplementary analysis (see reply to reviewer 1).

Since SCA is a new and complex type of analysis, a few additional sentences explaining it – and especially explaining how to interpret the results – would be very useful. The comparisons with Orben & Przybylski are very useful, so thank you for that. I still find it difficult to interpret the figure especially.

We have added additional information about the analytical approach [line 352-360] as well as two sentences to the caption of Figure 2 to help readers to interpret the SCA.

Third, there are some surprising inconsistencies in responses to similar items as illustrated in Table 4, e.g. Continuation despite problems. It seems especially strange that the endorsement of icd113 continuation is only 8.3% when this seems to be a much broader criterion than cvdat8 (endorsed by 23.3%). I notice that icd113 is missing something like "voor je", which is present in icd114. I also note that cvdat8 gives examples. I'm wondering what the authors think is the reason for this differential endorsement. In any case, it would be good to better understand what might account for the differences.

Thank you for calling attention to what is indeed an odd result. We do not have a good explanation for this—other similar items between the CVAT and ICD do not noticeably diverge and neither do we see lower endorsement rates across the board in the ICD measure (e.g., ICD loss of control was endorsed *more* frequently than CVAT loss of control, 9.7% vs 6.4%), so this may simply be (admittedly extreme) chance variation. Endorsement rates are presented as additional context for the reader but are not a core part of paper (especially given controversy about dichotomizing ordinal measures), so we prefer

not to speculate too much about this but will need to keep an eye out for these and related items in the future.

Also, I'm not sure if the chosen colors are easily understood by people with colorblindness-this might be something to consider as well.

We have used the viridis color palette

(<https://cran.r-project.org/web/packages/viridis/vignettes/intro-to-viridis.html>) for figures in which color conveys information, which has been designed to be colorblind-friendly.

Another small thing is that Orben and Przybylski is referenced a lot with "technology use"; it might be good to quickly mention in the intro what that study was and how they operationalized tech use

We have added a short description of this study where it first appears for context.

Abstract

Include sampling method

"identify a maximally parsimonious..." this phrase is somewhat hard to understand.

We have rephrased this sentence to be clearer.

Methods (by number)

168: People outside of Europe might not know what Flemish subjects are, so you could consider adding "Dutch-speaking people of Belgium" or some such.

We have added information to explain this.

Also, please consider describing more about the sampling procedure, websites, who might visit them, etc., the total number of survey questions, and whether survey questions were randomized.

We have added additional information about the total length of the questionnaire and the sampling procedure (see also reply to rmajor point above).

179: Average time on the DataVerse page differs from median time by almost 800 minutes; is this correct?

Fixed, see above replies to reviewers 1 and 2.

190: Were any other aspects compared, e.g. where the respondent was recruited from?

(see reply to major point above) The additional tests for differences between Dutch and Flemish participants are now reported in the Participants and Cleaning section [line.

201: Consider adding that reliability measures for each scale appear in Table 4.

The results in table 4 are based on single-item predictors of well-being, and therefore reliability cannot be computed for them. We have added reliability for all non-dysregulated gaming measures to Table 1 (see reply to reviewer 2 above), while reliability for the dysregulated gaming measures from which these single items come continue to appear in Table 2.

Dysregulated gaming: Consider adding averages/average endorsement percentages for clinical populations where available just to give context for the sample's average/endorsement percentages.

This would be useful information, but we believe it to be beyond the scope of this paper. Endorsement rates are provided for context only and should be viewed with the caveat that dichotomizing continuous/ordinal measures, especially when this may have diagnostic implications, is controversial. A rigorous comparison between clinical and non-clinical populations' response patterns would be a valuable topic for future work.

246: Consider adding year to "the time that this study was conceptualized". Might be worth discussing how the criteria used compare to what was eventually adopted, either here or in Discussion.

We have clarified the year here and the fact that the ICD-11 criteria have not changed in the interim [line 260].

270: Please add a link or references to the ODBA on OSF.

We have added a link to the ODBA's page on the OSF [footnote 1].

283-4: Please explain why the final item is divergent.

The final ODBA item assesses one of the definition's exclusion criteria, namely that a behavior should not be conceptualized as behavioral addiction if "the functional impairment results from an activity that, although potentially harmful, is the consequence of a willful choice". We have clarified the basis for including this item and also highlight opportunities for operationalizing the remaining ODBA exclusion criteria [line 292].

331: I could not find the exact numbers in the supplementary materials because I saw only the DataVerse referenced in the table note; perhaps insert the OSF link here.

We have reformatted this table to accommodate the numbers in the figure directly (see above).

334: Please add 1-2 more sentences that describe SCA, its assumptions and how it treats missing data. Also, are there corrects for multiple comparisons, or is this not necessary in this approach?

We have added additional information on these topics (see reply to reviewer 2 above).

372: Why are combinations of covariates not part of the specifications? Is this not possible?

With 16 covariates, including all possible combinations of all covariates would have increased the number of models from 972 to over 66,000. In order to keep results intelligible and not overemphasize more peripheral components of the model (given that the focus was analytical flexibility in the context of dysregulated gaming and well-being measures), we adopted a simpler approach. We welcome follow-up work, using our open data or other data, to see if we may have missed some moderation effect within the covariates—given how little influence they had on the overall results in the curve, however, we suspect that this is of negligible importance.

Results (by line number)

398: How does McDonald's omega compare to alpha? Is a "good" value also $>.8$?

We have added a sentence to clarify that readers can interpret omega values similarly to alpha, using whichever benchmarks they prefer.

423: VAT was fine as a single factor when it was developed, right? Some explanation of why it doesn't seem to hold together in this sample might be good.

One likely factor is the presence of correlated residuals in the original validation study—for interpretability and consistency with the other measures, we do not use the factor model from the original study our analyses. Previously, we mentioned this in the limitations section [line 711 of original manuscript], but we have moved this into the factor analytic results [line 425 of revised manuscript] to make it easier for the reader.

445: What are the possible reasons for the low association between ODBA and wellbeing? Could this be related to the divergent item wording?

The ODBA item assessing one of the definition's exclusion criteria—"the time you spend on games is a conscious choice"—appears indeed to be the primary cause of the low association, exhibiting low correlations with well-being by itself and loading poorly onto its hypothesized factor in the CFA. We discuss this in the paragraph beginning on line 579 [line 587 of revised manuscript]. Given that this scale was drafted on an exploratory basis and because of the consistent pattern of results found for

the other 5 scales/8 operationalizations, we prefer not to belabour this point too heavily in the manuscript. We look forward to future work investigating all of the proposed exclusion criteria, and to improvements/operationalizations associated with the ongoing ODBA project.

Figure 2: This could use a bit more explanation. What is being regressed on what? Do the lines correspond to points in Orben & Przybylski? I'm still not sure what the models are ranked by. I would also suggest using "control variables" rather than "controls".

We have added additional information about the interpretation of this figure in its caption (see also replies above), and have relabeled the control variables as suggested.

Table 3: Please include outcome and predictor variables or concepts (e.g., regressions of dysregulated gaming on wellbeing) in the table title and/or notes.

We have retitled this table to be more clear about its contents, as well as provided more details in the caption.

464: Do you have a citation for the decomposed variance approach? Or is it part of the original SCA paper?

This does not yet appear to be a widely-used technique for specification curves, but was inspired by the specr vignettes and intraclass coefficients for multilevel models, which are now both referenced and explained in slightly more detail [line 468].

474: What a huge change! I realize this is in the discussion, but something like "suggesting...." would be good here so it can be immediately interpreted.

We have now given some context to this so that readers have an idea of how to interpret it in the results section itself. [line 484]

Figure 3: "Value" is not easily interpreted as an axis title. I'm not sure how Fig 3 relates to Fig 2. It seems like perhaps the distributions are not for each analysis choice but for all analyses in which they appear?

These two are equivalent—each box plot in Figure 3 is essentially a summary of its respective bar in the bottom half of figure 2 that is easier to interpret. We have rewritten the caption of Figure 3 to be clearer and better reflect this correspondence.

Fig 4: Perhaps add briefly why ODBA is excluded

We have added a brief description to the caption to explain this.

492: for the explanatory variable of...? Include concept here

We have clarified this.

Table 4: The difference in effect sizes between similar items is surprising, e.g. Continuation despite problems. I see the percent endorsing these items is also different. This might benefit from further explanation and inclusion of percent endorsing the item.

The difference in effect sizes between these two similar items is indeed surprising. The item-level analyses are included on an exploratory basis, and given that the confidence intervals of the (median estimates) of the two continuation items overlap, we prefer not to speculate too much based on subtle differences in wording. Nonetheless, it will be important to eventually establish using e.g. cognitive pretests how slight changes in wording can change response patterns/what participants perceive to be the construct of interest.

Discussion (by line number)

How could the fact that all models yielded significant results relate to the different subgroups of the sample or the recruitment method?

We found no evidence for subgroups (outliers, country of origin) meaningfully changing the pattern of significant results. As for the recruitment method, we prefer not to speculate too much on this either: our sample is typical for survey studies in this field, including engaged gamers with low rates of dysregulation. Overall, the data indicate that the relationship is robust with relatively large effects (hence high power and exclusively significant results).

510: Typo-"flexibility's" flexibilities

Fixed.

557: Did Gentile even measure depression and anxiety as possible causes of PG in their longitudinal model? This is a problem with the conceptualization of causality in this body of literature and could be mentioned.

Although they mention the possibility, the Gentile paper does not report any effects of depression/anxiety on subsequent dysregulated gaming, only dysregulated gaming leading to depression/anxiety. While we do not want to diverge too far in what is already a complicated and long paper, we have added a short note mentioning the necessity for better causal data and inferences.

641: Why might the full versions not be unidimensional? Wouldn't this have appeared in prior research? It might be good to clarify what might be different here.

We have clarified that one source of the discrepancy is the presence of correlated residuals (see also response above). An additional difference is that the VAT was originally validated on a sample of high school adolescents, whereas the current sample is 16 and older, with a mean of 24 years old.

652: "in unique ways" is not very clear. Your earlier suggestion was that there might be a confounding factor causing gaming for escape, right?

We have altered this sentence to more specifically reflect our assertion that escapism may be confounded.

728: Please discuss the weaknesses and assumptions of the approaches, and items with the same concept (e.g., continuation despite problems) might receive such different endorsements from 2 scales.

We have added an additional sentence to give context to this topic (see also reply above)

735: "was attributable to the choice of dysregulated gaming measure" – wasn't this once ODBA was dropped?

We have clarified that this refers to the choice among remaining dysregulated gaming measures after excluding ODBA.

738: missing "to" in "noise the association"

We have fixed this typo.

Appendix

How were the items translated to English? I wonder whether the translations from English ICD-11 to Dutch might be responsible for the difference between the similar Table 4 items and how they were endorsed here.

Here are some native-English speaker/Dutch language learner suggestions that could be considered, either as things to explain some of the surprising findings or for future use:

VAT vs CVAT mood modification/withdrawal: "rot" is defined differently in Dutch and English between the questions that use it (or don't use it, i.e. cvidat2 Withdrawal, which specifies the feelings clearly)

*vat14: English has plural "problems" while Dutch has "a problem"; civdat5 does have the plural
vat4: English translation is missing "often"; without that there is a sense of "always" or "never" prefer,
perhaps*

cvidat2: English wording adds "were not allowed"

*cvidat8: "You played games" doesn't convey the same flavor of continuing as "je hebt toch games
gespeeld"*

cvidat9: gaming appears twice in the English translation

icd112: I don't think "other life interests" translates to "hobby's", but is meant to include all daily activities including functional roles like worker, student, family member.

icd113: Does the Dutch imply "for you"?

icd114: English "problematic" is different from Dutch "creating problems". I would suggest the difference between the wording of this question and the endorsement of a long syndrome of symptoms and problems be brought out in the Discussion. (e.g., it's not just that gaming is causing problems, but that the combination of symptoms and problems has persisted).

odba2: I'm not sure if the "om te minderen" could be misinterpreted as "problemen om te minderen" or if gaming is clearly implied

odba3: Why was this criteria divergent from what was in the OSF definition?

Thank you, these are all good catches/suggestions—in most cases, the Dutch wordings are the canonical ones, while English wordings are simply intended as a useful reference for the reader and are not the result of a formal translation process (the GKO is an exception to this, and translations for the VAT and CVAT2 are informed by previous translations appearing in their validation studies but have been slightly altered where we felt it better reflected the Dutch wording). We have made sure to clarify this for each scale in the notes of their respective appendix tables, and have also improved upon certain translations where appropriate based on additional discussion and the points raised here.

(Note: due to quirks with latexdiff and trying to make a tracked changes document, these changes are not currently highlighted. Slight changes were made to vat4, vat6, vat14, cvidat2, cvidat8, and icd113. The typo in cvidat9 was also corrected, and the English for the GKO items was also rephrased into second person to match the format of the other items.

Appendix B

Second Revision

We thank the reviewers again for their time to review the revised manuscript and are pleased that we were able to address the majority of their points of feedback.

Reviewer 1

Comments to the Author(s)

I think the authors adequately addressed the remaining issues raised by the reviewers and recommend publication.

Thank you for your time and positive recommendation.

Reviewer 3

Comments to the Author(s)

Thank you for attending to my multiple suggestions for revisions. I am satisfied with all of them. I do recommend some very minor things for clarity and to emphasize the impact of the findings.

If possible, could the many new findings and recommendations be listed in a highlights section or figure? There are so many implications for further research that I think this would be very useful.

Subject to it being allowed under the RSOS editorial policies, we agree that this would be a valuable addition to the paper, as we share concerns that the paper presents so many results that it can become difficult to follow. The revision includes a summary box in the introduction to focus the reader's attention throughout the manuscript.

p. 6

line 183 - the 87% is of what?

line 193 - sample size of 424 perhaps?

We have clarified the wording in these two sentences.

p. 7 line 261 - has the ADM been used in adults? How might the questions not apply?

We have added a sentence to say that while we could not find previous work that used this scale in adults, items address established depression symptoms not specific to adolescents, and the high reliability suggests that it performed acceptably in our sample.

p. 8 line 277 - contains one item - add "for each criterion"?

We have corrected this typo.

p. 14 Table 3 - I suggest you add something like "bivariate linear" to the "summary of regression coefficients" to the title

We have added some additional detail to the title of this table.

Figure 2 references dots, but I am unable to make out any dots in the figure. I think perhaps the top graph is simply too clustered, but I'm not sure.

We have clarified that the dots are indeed so dense as to appear like a line, and that readers can also note the darker color to indicate the position of the effect size estimates, with the lighter color corresponding to confidence intervals.

p. 19 line 680 - important implications like this and the many others could be highlighted in a box at the beginning of the paper

p. 20

line 707 - this is really important as well and could be highlighted

line 719 - could mention the need to explore confounding (and highlight)

line 735 - another important implication

These are now highlighted in the summary box (see reply above).

p. 85 Table A4-should the caption read the ODBA measure?

We have corrected this typo.

I would also like to suggest that should the authors pursue future studies with online recruiting, they should consider using non gaming-specific incentives such as bol.com or coolblue gift cards. Using a gaming-specific incentive could make more casual gamers less likely to participate.

Thank you for this suggestion, we will keep this in mind for future work!